# Hierarchical Multi-Scale Graph Neural Networks: Scalable Heterophilous Learning with Oversmoothing and Oversquashing Mitigation

**MD Sazzad Hossen** [1]   **Avimanyu Sahoo** [1]

## Abstract

Graphs with heterophily, where adjacent nodes carry different labels, are prevalent in real-world applications, from social networks to molecular interactions. However, existing spectral Graph Neural Network (GNN) approaches tailored for heterophilous graph classification suffer from hub-dominated (node with large degree) aggregation and oversmoothing, as their suboptimal polynomial filters introduce approximation errors and blend distant signals. To address the degree-biased aggregation and suboptimal polynomial filtering, we introduce a Hierarchical Multiview HAAR (HMH), a novel spectral graph-learning framework that scales in near-linear time. HMH first learns feature- and structure-aware *signed* affinities via a heterophily-aware encoder, then constructs a soft graph hierarchy guided by these embeddings. At each hierarchical level, HMH constructs a sparse, orthonormal, and locality-aware Haar basis to apply learnable spectral filters in the frequency domain. Finally, skip-connection unpooling layers combine outputs from all hierarchical levels back into the original graph, effectively preventing hub domination and long-range signal bottleneck (oversquashing). Experimentation shows that HMH outperforms state-of-the-art spectral baselines, achieving up to a 3% improvement on node classification and 7% points on graph classification datasets, all while maintaining linear scalability. Our code is available at https://github.com/sazzad1008/Scalable-HMH101

## 1. Introduction

Real-world graphs are rarely uniform: some regions are relatively homogeneous with nearby nodes sharing similar features and connectivity, while others are heterophilous, where neighbors differ sharply in both attributes and structure. In graph learning, homogeneous areas typically benefit from smoothing/averaging (low-pass filtering), whereas heterophellous regions require contrast amplification (high-pass filtering) (Chien et al., 2021). Spectral Graph Neural Networks (GNNs) aim to capture such signals by projecting onto a global eigenbasis of the graph Laplacian or its polynomial approximation (Zhu & Koniusz, 2021). Using a global basis, whether it is explicitly precomputed (He et al., 2022) or implicitly approximated using high-order polynomials (Huang et al., 2024b), often sacrifices graph locality. The induced filters can spread energy across many hops and mix signals from semantically unrelated regions, which may blur fine-grained structural distinctions among small clusters that are especially important under heterophily (He et al., 2022).

In addition, conventional polynomial bases (e.g , Chebyshev) are limited by the conditioning of polynomial parameterizations on graphs. In the theoretical setting, Chebyshev polynomials are orthogonal on $[-1, 1]$ under a *continuous* weight function $(w(x) = 1/\sqrt{1 - x^2})$. Although graph filters rescale the Laplacian spectrum to $[-1, 1]$, a real graph does not provide this continuous weighting since it is governed by a *discrete* spectral measure (the rescaled eigenvalues and their multiplicities), leading to leakage across frequency channels and reduced effective expressivity of the filter bank (Guo & Wei, 2023). Moreover, high-order polynomial approximations can be prone to error and the resulting bases are typically *static* and not adaptive to local feature/structure heterophily (Huang et al., 2024b; Zheng, 2024).

On the other hand, spatial GNNs filter graph signals directly by employing neighborhood aggregation and often exhibit an inherent low-pass bias (Zhu & Koniusz, 2021). To address this, recently proposed signed message-passing methods (SMPs) incorporate edge signs to learn the contrast between the node signals. SMP-based models suffer from sign cancellation across depth (e.g., every two layers), progres-

[1]University of Alabama in Huntsville, Huntsville, AL, USA. Correspondence to: MD Sazzad Hossen <mh0255@uah.edu>.

*Proceedings of the 43rd International Conference on Machine Learning*, Seoul, South Korea. PMLR 306, 2026. Copyright 2026 by the author(s).

sively erasing heterophilous contrasts (Liang et al., 2024). To prevent signal-signed flipping, several spatial models employ Chunked Message Passing (CMP) approaches, which bucketize neighbors by similarity and subsequently consolidate all chunks/buckets via a global aggregation step (Pei et al., 2024). Both approaches mix signals across graph regions. This mixing is particularly detrimental in degree-imbalanced graphs: densely connected, high-degree hub nodes can dominate message passing, eroding the signals of small (spoke) heterophilous clusters, leading to indistinguishable features (hub domination) and ultimately exacerbating signal oversmoothing (Keriven, 2022). These limitations motivate a filter that is (i) spectrally well-conditioned to prevent leakage across frequency channels, (ii) localized to avoid long-range contamination, and (iii) learns high-pass and low-pass filters without sign cancellation pathology.

To address the existing limitations, we propose a novel model, referred to as HMH. The HMH first introduces an adaptive heterophilous encoder that assigns positive weights to homophilous edges and persistent negative weights to heterophilous edges. The encoder is also designed to serve as an adaptive high-pass and low-pass filter, thereby preserving the high-frequency contrasts that conventional static methods typically fail. The encoder also incorporates structural similarity and feature affinities to generate per-node scores that facilitate hierarchical clustering (coarsening). At each hierarchical level, the encoder is used to construct the Haar basis, which forms a sparse, orthonormal eigenbasis explicitly capable of capturing localized low-frequency (within-cluster) and high-frequency (between-cluster) variations. Diagonal spectral filtering in this basis selectively amplifies heterophilous channels and attenuates hub-dominated signals. Skip-connected unpooling is employed to reintegrate filtered signals from each level into the original nodes, enriching them with multi-scale context. The coarsening hierarchy reduces effective path lengths logarithmically, allowing gradients and information to propagate without the exponential decay caused by oversquashing (Di Giovanni et al., 2023). Finally, we leverage the hierarchical tree to perform graph pooling for graph classification, reducing the pooling overhead from quadratic $O(n^2)$ in prior methods to near-linear $O(n)$ time (Li et al., 2024).

The contributions of the paper are: (i) We introduce a label-free adaptive signed affinity that prevents sign cancellation. (ii) We introduce the first framework that constructs an orthonormal, multi-scale, sparse spectral basis in near-linear time, scaling effortlessly to large graphs; (iii) We rigorously prove and empirically confirm that conventional GNNs suffer from **hub domination**, **oversmoothing**, and **oversquashing**, whereas our proposed **HMH** overcomes these pathologies; and (iv) we demonstrate that HMH achieves state-of-the-art (SOTA) accuracy on both node and graph classification tasks while maintaining linear scalability.

**Conflict of Interest Disclosure.** The authors declare no financial conflicts of interest related to this work.

## 2. Related works

**Spectral Filtering:** Early models, like ChebNet (He et al., 2021), used truncated Chebyshev polynomials to approximate the eigenbasis. Conversely, Generalized Page Rank GNN (GPR-GNN) (Chien et al., 2021) utilizes generalized PageRank weights with monomial bases for approximations. In contrast, BernNet (He et al., 2021) and JacobiConv (Wang & Zhang, 2022) provide Bernstein and Jacobi polynomials, respectively, to enhance the interpretability and adaptability of the bases. ChebNetII (He et al., 2022) addressed the issue of Chebyshev polynomials overfitting by employing interpolation, whereas OptBasisGNN (Guo & Wei, 2023) aimed to make basis polynomials orthogonal to accelerate convergence. The above methodologies continue to employ dense, fixed graph-wide bases that prevent global message mixing.

**Hierarchical Graph Learning:** Learnable pooling methods, such as DiffPool (Ying et al., 2018), SAGPool (Lee et al., 2019), TopKPool (Diehl, 2019), utilize node-scoring to coarsen homophilous graphs at a singular resolution incurring $O(n^2)$ computational cost (Li et al., 2024). EigenPool (Ma et al., 2019) and Haar-based pooling employ an expensive eigendecomposition to coarsen the graph, incurring an $O(n^3)$ computational cost, which constrains scalability and overlooks local heterophily (Wang et al., 2020).

## 3. Preliminaries

Let $G = (V, E)$ be a simple, undirected graph, where $V$ denotes the set of vertices and $E$ denotes the set of edges. Its adjacency matrix $A \in \mathbb{R}^{n \times n}$ has entries $A_{ij} = 1$ if $i, j \in E$ and 0 otherwise, and $D = \text{diag}(A\mathbf{1})$ is the degree matrix, where $\mathbf{1}$ is the all-ones vector. Given node features $X \in \mathbb{R}^{m \times d}$ and normalized Laplacian $\mathcal{L} = U\Lambda U^\top$, a spectral layer rescales graph Fourier components by $g(\mathcal{L})X = U\,g(\Lambda)\,U^\top X$, where $g$ is spectral filter. The polynomial spectral methods implements $g$ with a polynomial, $g(L) \approx \sum_{r=0}^{R} \theta_r L^r$, where $R \in \mathbb{N}$ is the polynomial order. With $k$ layer message–passing, a GNN learns the graph signal as a polynomial through a propagation operator $P$, which essentially induces the global eigenbasis of the propagation operator $P$ given by

$$H^{(k)} \approx \sum_{r=0}^{k} \Theta_r\, P^r\, X = g_P(P)\, X\ = U_P\, g_P(\Lambda_P)\, U_P^\top X,$$

(1)

where $U_P \Lambda_P U_P^\top$ is the eigen-decomposition of $P$ and $\Theta_r$ are learnable linear parameters

Real-world graphs often exhibit different levels of homophily in different regions of the graph. We quantify

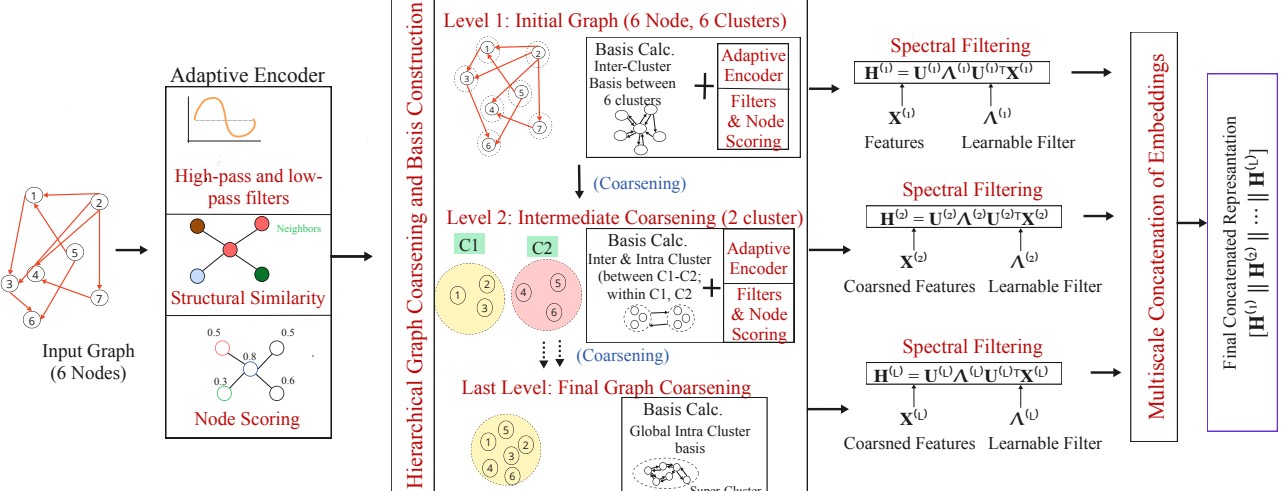

*Figure 1.* Overview of the proposed HMH framework. The process begins with the input graph and node features, followed by adaptive heterophilous encoding and hierarchical graph coarsening. At each level, class-aware Haar bases are constructed, and diagonal spectral filtering is performed. Finally, multi-scale skip-connected fusion aggregates information from all levels.

*homophily* as $H_{\text{lab}} = \frac{1}{|E|} \sum_{(u,v)\in E}[y_u = y_v]$, where $y$ is the node label. To address the discrepancy between nodes homophily, the *sign-based message passing (SMP)* (Liang et al., 2024; Chien et al., 2021) algorithms introduced signed adjacency matrix, $S$, defined as $S_{uv} = +1$ if $(u,v) \in E$ and $y_u = y_v$, $S_{uv} = -1$ if $(u,v) \in E$ and $y_u \neq y_v$. So the feature of each layer is updated as, $H^{(k+1)} = \sigma\big(S H^{(k)} W^{(k)}\big)$, $H^{(0)} = X$, where $S$ works as a propagation operator and $W$ is weight matrix. Using (1), the $k$-layer SMP can be expressed as a spectral filter in the global basis of $S$. To solve the sign flipping of SMP, *chunked multi-track aggregation (CMA)-based methods* (Pei et al., 2024) propose to split each node's neighbors into $t$ tracks (e.g., homo/hetero) with learned attention $s_{ij,t}$ and aggregate messages $m_{i,t}$ per track $t$. Vanilla GCNs are known to exhibit two depth-related pathologies. *(i) Oversmoothing* occurs as the number of layers grows and all node embeddings converge to their class means (Epping et al., 2024). *(ii) Oversquashing* (Topping et al., 2021), which occurs when gradients (or messages) from distant nodes decay exponentially with graph distance, preventing long-range information flow. While many heterophilous GNNs aim to address these issues, their limitations under hub-dominated graphs and suboptimal bases remain, as discussed next.

## 4. Limitations of Existing Models

We highlight two failure modes in existing graph learning models that together explain the poor performance on hub-dominated graphs (Luan et al., 2022; Guo & Wei, 2023).
**Limitation 1: Hub Domination.** Consider two small clusters $A$ and $B$ densely connected to a dominant Hub region $H'$ (either a single high-degree node or a homophilous clus-

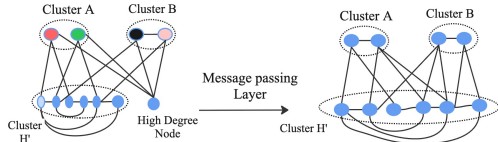

*Figure 2.* Hub aliasing phenomenon in cluster $(A, B, H')$.

ter of high-degree nodes). Due to $H'$'s significantly higher degree relative to the spokes (small cluster of low degree nodes), the resulting higher signal contribution statistically overwhelms the local features of $A$ and $B$ during message passing. Despite $A$ and $B$ having distinct class labels, this "Hub Domination" causes the embeddings to drift toward a common hub-induced representation, erasing the decision boundary between them. The following theorem establishes the theoretical basis of hub domination, and empirical results and analyses are presented later in Table 3.

**Theorem 4.1.** *The representational distance between any node in $A$ and any node in $B$ decays exponentially after multiple layers of SMP or CMA. (Proof in Appendix A)*

**Limitation 2: Suboptimal basis.** Fixed polynomial bases (e.g., Jacobi, Chebyshev) assume continuous orthogonality, which does not hold on the discrete spectrum of irregular graphs. Consequently, polynomial basis become correlated, hindering the model's ability to decouple frequency components and adapt to local graph structures.

## 5. Methodology

In this section, we present the proposed HMH spectral graph-learning framework. HMH performs multi-resolution anal-

ysis of graph signals, capturing both local interactions and global structural trends, as shown in Fig. 1. The pipeline begins with a heterophily-aware encoder that computes node scores from local features and structural cues. These scores guide a clustering step that forms the supernodes for the next level. We apply the same encoder–clustering procedure recursively at every level to construct the full hierarchy. At each level, we build a level-specific spectral basis $U^{(\ell)}$, enabling diagonal filtering in the transformed domain and multi-scale fusion back to the original graph. We detail each component below.

### 5.1. Heterophilous Encoder

The goal is to introduce an adaptive encoder that generates node scores based on feature and structural similarity, guiding the graph coarsening process. In addition, the encoder should act as a learnable high-pass and low-pass filter by integrating the feature affinities and structural patterns to determine node similarity. We follow a two-step process:

(i) **Adaptive Similarity Computation.** At layer $k$, let the node embeddings be $H^{(k)} \in \mathbb{R}^{n \times d_k}$ and $h_i$ is the $i$th node embedding. For each neighbor $j \in \mathcal{N}(i)$, we compute (I) **feature affinity** $S_{\text{att}}^{(k)}(i,j) = \sigma(\mathbf{w}_{\text{att}}^{\top}[h_i^{(k)} \| h_j^{(k)}])$, where $\mathbf{w}_{\text{att}}$ is learnable weights and ; and (II) **structural similarity** $S_{\text{struct}}^{(k)}(i,j) = \frac{|\mathcal{N}_2(i) \cap \mathcal{N}_2(j)|}{|\mathcal{N}_2(i) \cup \mathcal{N}_2(j)|}$, where we use two-hop neighbors $\mathcal{N}_\in$ to find similarity using shared neighbors beyond direct adjacency. We integrate scores to form a normalized similarity matrix $\widetilde{S}^{(k)}$, where each

$$\widetilde{S}_{ij}^{(k)} = \text{softmax}_{j \in \mathcal{N}(i)}\left(S_{\text{att}}^{(k)}(i,j) + S_{\text{struct}}^{(k)}(i,j)\right). \quad (2)$$

(ii) **Adaptive Signed Adjacency & Propagation.** To capture both homophily and heterophily, we transform the binary adjacency $A$ into a signed, continuous matrix $A_{\text{adp}}^{(k)}$. We assign positive weights to similar nodes and negative weights to dissimilar ones by computing

$$A_{\text{adp}}^{(k)} = \left(2\widetilde{S}^{(k)} - \mathbf{1}\mathbf{1}^{\top}\right) \odot A^{(k)} = \left(2\widetilde{S}^{(k)} - 1\right) \odot A^{(k)}. \quad (3)$$

where $\odot$ is element-wise product. Alternatively, we set $(A_{\text{adp}}^{(k)})_{i,j} = 2\widetilde{S}_{ij}^{(k)} - 1$ if $A_{ij} = 1$, and 0 otherwise. This creates a dynamic filtering mechanism. That is, when two nodes are highly similar (e.g., $\widetilde{S} \approx 0.9$), the resulting edge weight is positive (about 0.9) and message passing behaves like a low-pass operator by averaging neighbor features. In contrast, low similarity (e.g., $\widetilde{S} \approx 0.4$) yields a negative weight, which emphasizes differences between neighbors and thus behaves like a high-pass (sharpening) operator. Using this graph-agnostic weighting, the encoder updates the feature as

$$Z^{(k+1)} = \sigma\left(A_{\text{adp}}^{(k)} Z^{(k)} W^{(k)}\right). \quad (4)$$

For every layer $k$, we recalculate the $\widetilde{S}^{(k)}$, thereby updating $A_{\text{adp}}^{(k)}$ according to (3), which mitigates the sign flipping and serves as an adaptive filter demonstrated in the following theorems.

**Theorem 5.1.** *Updating $A_{adap}$ using (3) with (2) at every propagation layer $k$ mitigates the periodic sign-flipping limitation of SMP. (Proof in Appendix ??)*

**Theorem 5.2.** *At every layer $\ell$ of message passing, the $A_{adp}$ acts as an adaptive combination of low-pass and high-pass filter. (Proof in Appendix B)*

### 5.2. Haar-Tree Formulation

To construct the hierarchy, we utilize the encoder embeddings $Z^{(\ell)} = \text{Encoder}(X^{(\ell)}, A_{\text{adp}}^{(\ell)})$ as node scores. These scores guide the partitioning of nodes into clusters and the subsequent aggregation of features and edges for the formed cluster via three specific steps:

(i) **Cluster Prototype:** To coarsen the tree at a predefined ratio $R$, we set the target number of nodes for the coarsened trees as $K_\ell = \max\left(1, \lfloor |V^{(\ell)}| \cdot R \rfloor\right)$. We employ k-means++ (Bahmani et al., 2012) on $Z^{(\ell)}$ to obtain $K_\ell$ the number of centroids $\{p_k^{(\ell)}\}_{k=1}^{K_\ell} \subset \mathbb{R}^{d'}$, which serves as prototypes of cluster $\{C_k^{(\ell)}\}_{k=1}^{K_\ell}$.

(ii) **Weighted Soft Assignment:** Rather than assigning each node to a single cluster, we use a probabilistic (soft) membership over all prototypes. For each node $i \in V^{(\ell)}$ and prototype $p_k^{(\ell)}$, we compute an affinity score $\Omega_{ik}^{(\ell)} = \langle Z_i^{(\ell)}, p_k^{(\ell)} \rangle$, where $k = 1, \ldots, K_\ell$, where $\langle ., . \rangle$ denotes inner product. To control the sharpness of the decision boundaries, we compute the row-stochastic assignment matrix as $(A_s^{(\ell)})_{ik} = \text{softmax}_k\left(\frac{\Omega_{ik}}{\tau}\right)$, where $\tau > 0$ is a temperature parameter that controls the sharpness.

(iii) **Feature and Edge Aggregation:** Given the node-feature $X^{(\ell)} \in \mathbb{R}^{|V^{(\ell)}| \times d_\ell}$ and the adjacency $A^{(\ell)} \in \{0,1\}^{|V^{(\ell)}| \times |V^{(\ell)}|}$ at level $\ell$, we obtain the next-level $(\ell+1)$ coarsened graph and its adjacency as

$$X^{(\ell+1)} = \left(A_s^{(\ell)}\right)^{\top} X^{(\ell)} \in \mathbb{R}^{K_\ell \times d_\ell}, \quad (5)$$

$$A_{kk'}^{(\ell+1)} = \sum_{t \in C_k^{(\ell)}} \sum_{q \in C_{k'}^{(\ell)}} (A)_{tq}^{(\ell)} \in \mathbb{R}^{K_\ell \times K_\ell}, \quad (6)$$

where, at level $\ell$, let $k, k' \in \{1, \ldots, K_\ell\}$ index the coarse (cluster) nodes, while $t, q \in V^{(\ell)}$ index the fine nodes. Thus, each coarse node's feature is the probability-weighted sum of its children's features, and two coarse nodes are adjacent whenever at least one edge exists between their constituent nodes at level $\ell$. Non-zero entries $(A_s^{(\ell)})_{ik} > 0$ means that a fine node $i$ contributes to a coarse node $k$ at level $\ell + 1$. Collecting these indices across levels yields the parent–child links of the Haar tree. The coarsening loop repeats until

$|V^{(L)}| \le h_t$, where $h_t$ is a user-specified threshold. Next, we discuss the algorithms to compute the class-aware Haar basis $U^\ell$ from the Haar tree.

### 5.3. Class-aware Haar Bases

The coarsening builds a hierarchy $\{G^{(\ell)}\}_{\ell=0}^L$ by iteratively merging nodes from the input graph ($L = 0$) to the coarsest level $L$ utilizing the encoder, which may collapse into a single supernode depending on the ratio $R$. At each level L, supernodes represent clusters from the previous level. We construct a spectral basis $U^{(\ell)}$ that decomposes signals into inter-cluster contrasts and intra-cluster details, enabling level-specific low-/high-pass filtering. The bases are built bottom-up: we construct the coarsest basis (intra-cluster only if $K_L = 1$) and recursively lift it to finer levels to preserve global structure while adding localized detail.

**Coarsest level ($\ell = L$).** Let the coarsest graph $G^{(L)}$ consist of $K_L = |V^{(L)}|$ supernodes. We construct $U^{(L)} \in \mathbb{R}^{K_L \times K_L}$ from a global scaling vector and $K_L - 1$ inter-supernode wavelets (if $K_L > 1$).

**(1) Global scaling vector.** To capture the global low-frequency (mean) component, we can define the basis as

$$u_{\text{sc}}^{(L)} = \frac{1}{\sqrt{K_L}}[1, 1, \dots, 1]^\top.$$

**(2) Inter-supernode (cluster) wavelets (coarse contrasts).** If $K_L > 1$, we need basis vectors to capture the differences *between* these supernodes. We construct $K_L - 1$ orthogonal wavelets using a standard recursive difference method. For $q = 1, \dots, K_L - 1$, the $q$-th wavelet contrasts the $q$-th supernode against the average of all subsequent supernodes:

$$w_{\text{coarse};\,q}^{(L)} = \sqrt{\frac{K_L - q + 1}{K_L - q}} \left( e_q - \frac{1}{K_L - q} \sum_{t=q+1}^{K_L} e_t \right),$$

(7)

where $e_q$ is the standard basis vector at index $q$ (a one-hot vector with $(e_q)_q = 1$ and zeros elsewhere). These vectors have zero mean, unit norm, and are mutually orthogonal. Hence $U^{(L)} = [u_{\text{sc}}^{(L)} \mid \cdots \mid w_{\text{coarse};\,K_L-1}^{(L)}]$ is orthonormal.

**Recursive lifting to finer levels ($\ell = L-1, \dots, 0$).** At level $\ell$, fine nodes $V^{(\ell)}$ are softly assigned to $K_{\ell+1} = |V^{(\ell+1)}|$ clusters through $A_s^{(\ell)} \in \mathbb{R}^{|V^{(\ell)}| \times K_{\ell+1}}$. For basis construction, we use the soft assignment matrix $A_s^{(\ell)}$ to define weighted cluster memberships. Feature coarsening, edge coarsening, and Haar filtering are implemented using the soft assignment weights.

**(1) Lift the global scaling vector.** To ensure that each node inherits the coarse-level average of its parent cluster, given $u_{\text{sc}}^{(\ell+1)} \in \mathbb{R}^{K_{\ell+1}}$, we lift the mean to the finer graph as

$$u_{\text{sc}}^{(\ell)} = A_s^{(\ell)} u_{\text{sc}}^{(\ell+1)} \in \mathbb{R}^{|V^{(\ell)}|}. \tag{8}$$

**(2) Inter-cluster wavelets (between clusters).** Treating the $K_\ell$ clusters as "supernodes," we construct $K_\ell - 1$ orthogonal wavelets that capture differences *between* clusters using the same procedure as in (7).

**(3) Intra-cluster wavelets (within a cluster).** Consider a cluster $C_k^{(\ell)}$ with $n_k$ nodes. We construct $n_k - 1$ strictly local wavelets to capture variations within this cluster. We iterate through the cluster's nodes and define "split points." For a split at index $r$, we define a wavelet $w_{\text{intra};k,r}^{(\ell)}$ that contrasts the first $r$ nodes (Left (c) group ) against the remaining $n_k - r$ nodes (Right (e) group) as

$$w_{\text{intra};k,r}^{(\ell)}(i) = \begin{cases} \sqrt{\dfrac{\beta_{k,r}}{\alpha_{k,r}(\alpha_{k,r} + \beta_{k,r})}}\, (A_s^{(\ell)})_{i,k}, & i \in \text{c}, \\[2ex] -\sqrt{\dfrac{\alpha_{k,r}}{\beta_{k,r}(\alpha_{k,r} + \beta_{k,r})}}\, (A_s^{(\ell)})_{i,k}, & i \in \text{e}. \end{cases}$$

(9)

with $w_{\text{intra};k,r}^{(\ell)}(i) = 0$ for $i \notin C_k^{(\ell)}$, where $\alpha_{k,r}^{(\ell)} = \sum_{j=1}^r (A_s^{(\ell)})_{i,k}$ and $\beta_{k,r}^{(\ell)} = \sum_{j=r+1}^{n_k} (A_s^{(\ell)})_{i,k}$ are the total membership masses of the Left and Right groups, respectively.

**Assembly and filtering.** Finally, we concatenate the scaling vector, inter-cluster wavelets, and intra-cluster wavelets to form the Haar basis as

$$U^{(\ell)} = \left[ u_{\text{sc}}^{(\ell)} \mid W_{\text{inter}}^{(\ell)} \mid W_{\text{intra}}^{(\ell)} \right] \in \mathbb{R}^{|V^{(\ell)}| \times |V^{(\ell)}|}.$$

The structure of the Haar basis is shown in Fig. 3. We filter features at level $\ell$ as

$$H^{(\ell)} = U^{(\ell)} \Lambda^{(\ell)} U^{(\ell)\top} X^{(\ell)}, \tag{10}$$

where $\Lambda^{(\ell)}$ is a learnable diagonal gain matrix controlling low-/high-frequency channels. Because $U^{(\ell)}$ is sparse and locally supported, both basis construction and filtering can be implemented in near-linear time on sparse graphs.

### 5.4. Prediction Task

For node-level tasks, we compute each node's final embedding by additive unpooling $\widehat{H}_j = H_j^{(0)} + \sum_{\ell=1}^L \sum_{i:j \in C_i^{(\ell)}} H_i^{(\ell)}$ so that every node carries its base feature plus the pooled summaries of all clusters across different scales it belongs to. For graph-level tasks, we coarsen until the number of nodes is $|V^{(L)}| = 1$. We then use the embedding of this final supernode for graph classification. The total loss is defined as $L_{\text{total}} = L_{\text{CE}} - \lambda_{\text{div}}\, L_{\text{div}}$, where $L_{\text{CE}}$ is the standard cross-entropy and

$$L_{\text{div}} = -\sum_{\ell=0}^{L-1} \frac{1}{|V^{(\ell)}|} \sum_{i=1}^{|V^{(\ell)}|} \sum_{k=1}^{K_\ell} A_{s,ik}^{(\ell)} \log A_{s,ik}^{(\ell)},$$

that maximizes the entropy of each node's soft assignment $(A_s^{(\ell)})_{i:}$. Hyperparameters $\lambda_{\text{div}}$ and $\lambda_{\text{rec}}$ balance the auxiliary terms.

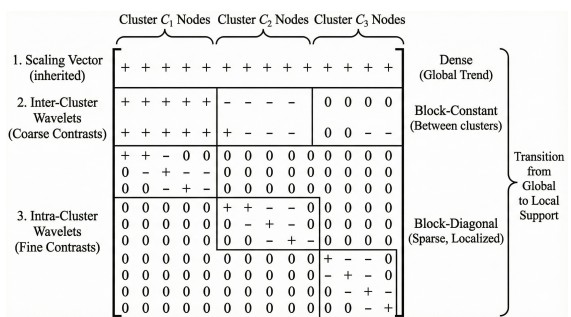

*Figure 3.* **Structure of the Haar basis matrix $U^{(\ell)}$. Columns are nodes ordered by assumed clusters** $(C_1, C_2, C_3)$

## 5.5. Theoretical Perspective

Figure 3 illustrates the distinct structure of the Haar basis matrix $U^{(\ell)}$ using a simplified example with three clusters $(C_1, C_2, C_3)$. The rows exhibit a hierarchical progression. The first row is a dense scaling vector that captures the global trend. The second and third rows correspond to inter-cluster wavelets, which are block-constant and encode contrasts between clusters. The remaining rows form intra-cluster wavelets, which are sparse (block-diagonal) and capture strictly local variations within each cluster.

This structure offers a decisive advantage over global polynomial bases, whose basis are dense and spread energy diffusely across distant nodes. By strictly confining the support of each wavelet, Haar filtering prevents information leakage between unrelated parts of the graph. Our localized basis preserves these high-frequency components without allowing the hub's dominant signal to overwhelm them, thereby mitigating hub domination. Furthermore, the strict orthogonality between the scaling and wavelet subspaces prevents frequency leakage and uncontrolled low-pass drift, reducing oversmoothing. The hierarchical nature of the construction logarithmically shortens communication paths to alleviate oversquashing. We formalize these theoretical guarantees in the following theorems.

**Theorem 5.3.** *A spectral filter based on orthonormal and local basis holds any region-specific signal pattern confined under filtering. (Proof in Appendix C)*

**Theorem 5.4.** *Let $A, B, H'$ post-filter centroids are $\mu_A, \mu_B, \mu'_H$ and set $\Delta_{AB} = \|\mu_A - \mu_B\|, \Delta_{AH'} = \|\mu_A - \mu'_H\|$, then $\frac{\Delta_{AB}}{\Delta_{AH'}} \geq \sqrt{\frac{M}{a+b}}\left(1 - \frac{2}{\sqrt{M}}\right) > 1$. Thus enlarging the hub $H$ can never shrink the spoke–spoke separation below the spoke–hub separation, avoiding hub domination. (Proof is in Appendix D)*

**Theorem 5.5.** *Regardless of the number of layers in the proposed HMH , it overcomes the problems of oversmoothing and oversquashing. (Proof is in Appendix E)*

## 6. Experiments

**Node Classification** We evaluate HMH on standard node-classification benchmarks spanning *homophilous* citation graphs (Cora, Citeseer, Pubmed, Coauthor-CS/Physics) and *heterophilous* graphs (Chameleon, Squirrel, Texas, ROMAN-EMPIRE, AMAZON-RATINGS, QUESTIONS, MINESWEEPER, TOLOKERS, etc.) (Platonov et al., 2023). Dataset statistics (nodes/edges, classes, homophily) are provided in Appendix G.1. For smaller datasets, we use a $60/20/20$ train/val/test split and report *inductive* performance, where test nodes are not used for supervision during training. For larger graphs, we follow the standard *transductive* protocol with validation tuning and early stopping, and report mean test performance over 10 splits (mean±95% CI). For the heterophily suite of Platonov et al. (2023), we use the official splits and metrics (ROC-AUC for MINESWEEPER/TOLOKERS/QUESTIONS; accuracy otherwise). Full training and tuning details are in Appendix F.

**Baselines.** We compare against representative methods from (i)polynomial filtering, (ii) heterophilly-specialized message passing, and (iii) spectral/transformer models: SIGN (Frasca et al., 2020), EvenNet (Lei et al., 2022), ChebNet II (He et al., 2022), BernNet (He et al., 2021), JacobiConv (Wang & Zhang, 2022), ARMA, UniFilter (Huang et al., 2024b), SLOG (Xu et al., 2024), LINKX (Lim et al., 2021), FAGCN (Bo et al., 2021), MTGCN (Reuter et al., 2025), M2M-GNN (Liang et al., 2024), and PolyFormer (Ma et al., 2024), TFE-GNN (Huang et al., 2024a).

**Results and discussion.** Table 1 summarizes node classification accuracy. Overall, **HMH achieves strong and consistent performance** on homophilic graphs, with **clear gains on challenging heterophilic benchmarks** and **competitive results at large scale**. On **homophilic** datasets (e.g., CS and Citeseer), where labels are dominated by smooth signals, the encoder learns mostly positive affinities enabling stable low-pass propagation, while Haar-domain diagonal gains suppress non-informative high-frequency components, improving denoising without oversmoothing. On **heterophilic** datasets (e.g., Minesweeper, Roman-empire, Squirrel/Chameleon, and Amazon-ratings), HMH improves over standard GNN baselines and remains competitive with specialized heterophilous methods by learning *signed, adaptive affinities* and selectively preserving contrastive (high-frequency) Haar components, which reduces destructive neighbor mixing and better maintains class boundaries. HMH also performs strongly on the large-scale **OGBN-Arxiv** dataset (169,343 nodes; 1,166,243 edges), indicating that hierarchical Haar filtering remains effective beyond small benchmarks. Additionally, the results from large-scale datasets are presented in Appendix F.1.

*Table 1.* Node classification performance (%). "—" indicates not reported under the corresponding suite/protocol.

| Method | Homophilic | | | Heterophilic and Large Scale | | | | | | | |
| | Citeseer | CS | Physics | Minesweeper | Tolokers | Roman-empire | Ogbn-arxiv | Questions | Squirrel | Chameleon | Amazon-ratings |
| --- | --- | --- | --- | --- | --- | --- | --- | --- | --- | --- | --- |
| GCN | 74.60 ± 1.00 | 90.50 ± 0.10 | 95.80 ± 0.10 | 78.80 ± 0.00 | 78.40 ± 0.10 | 53.74 ± 0.29 | 61.74 ± 0.29 | 97.00 ± 0.00 | 23.40 ± 0.70 | 27.20 ± 1.20 | 42.00 ± 0.20 |
| GAT | 73.90 ± 0.80 | 93.80 ± 0.10 | 95.80 ± 0.10 | 78.70 ± 0.10 | 77.60 ± 0.00 | 52.74 ± 0.29 | 67.60 ± 0.30 | 97.00 ± 0.00 | 24.60 ± 0.60 | 28.40 ± 2.00 | 48.20 ± 0.10 |
| GPRGNN | 75.60 ± 0.30 | 94.20 ± 0.10 | 96.60 ± 0.10 | 79.10 ± 0.00 | 77.50 ± 0.10 | 74.74 ± 0.29 | 68.40 ± 0.30 | 97.00 ± 0.00 | 34.30 ± 0.90 | 47.20 ± 2.00 | 41.40 ± 0.40 |
| BernNet | 74.50 ± 1.50 | 93.80 ± 0.10 | 95.90 ± 0.00 | 78.80 ± 0.00 | 77.20 ± 0.70 | 72.74 ± 0.29 | 68.74 ± 0.29 | 96.90 ± 0.10 | 36.10 ± 0.70 | 37.80 ± 0.70 | 39.80 ± 0.20 |
| ChebNetII | 74.00 ± 0.90 | 64.00 ± 0.10 | 95.80 ± 0.00 | 82.30 ± 0.10 | 78.30 ± 0.30 | 73.74 ± 0.29 | 71.12 ± 0.22 | 96.90 ± 0.10 | 35.00 ± 0.40 | 33.50 ± 0.50 | 39.30 ± 0.10 |
| GCNII | 74.80 ± 0.80 | 91.80 ± 0.10 | 96.10 ± 0.10 | 78.80 ± 0.00 | 77.80 ± 0.10 | 74.74 ± 0.29 | 69.38 ± 0.40 | 97.00 ± 0.00 | 31.40 ± 0.80 | 38.00 ± 1.00 | 42.90 ± 0.20 |
| JacobiConv | 55.90 ± 9.80 | 88.20 ± 0.80 | 92.40 ± 1.00 | 78.80 ± 0.00 | 70.40 ± 0.57 | 70.74 ± 0.29 | 68.38 ± 0.40 | 96.70 ± 17.60 | 22.10 ± 1.70 | 30.90 ± 1.50 | 35.50 ± 1.00 |
| Specformer | 81.69 ± 0.78 | 96.07 ± 0.10 | 97.70 ± 0.60* | 89.93 ± 0.41 | 80.42 ± 0.55 | 69.94 ± 0.34 | – | 96.49 ± 0.58* | 27.60 ± 1.70 | 28.20 ± 2.30 | 43.18 ± 0.60 |
| OptBasisGNN | 76.46 ± 1.60 | 93.80 ± 0.10 | 94.10 ± 0.60 | 84.80 ± 2.40 | 79.85 ± 1.63 | 68.34 ± 0.30 | 72.27 ± 0.40 | 97.20 ± .40 | 28.66 ± 1.10 | 39.70 ± 2.40 | 43.35 ± 1.34 |
| UniFilter[†] | 77.80 ± 0.90 | 92.80 ± 0.10 | 93.10 ± 0.30 | 86.80 ± 0.70 | 78.12 ± 0.83 | 74.50 ± 0.56 | - | 96.20 ± 0.45 | 33.10 ± 0.80 | 41.20 ± 1.30 | 42/43 ± 0.96 |
| TFE–GNN(sum) | 82.83 ± 1.24 | 96.10 ± 0.17 | 97.85 ± 0.13 | 86.85 ± 0.33 | 79.23 ± 1.13 | 74.20 ± 0.35 | 71.34 ± 0.55 | 96.20 ± 1.45 | 31.47 ± 1.15 | 41.16 ± 1.41 | 40.16 ± 1.81 |
| M2M-GNN[†] | 77.20 ± 1.80 | 90.35 ± 0.30 | 90.35 ± 0.60 | 82.25 ± 1.43 | 80.85 ± 1.13 | 68.36 ± 3.23 | 72.15 ± 0.83 | 95.39 ± 1.54 | 33.60 ± 1.70 | 32.20 ± 2.30 | **47.18 ± 0.60** |
| MTGCN | 76.91 ± 1.30 | 92.54 ± 0.45 | 94.72 ± 1.24 | 82.20 ± 1.21 | 77.20 ± 0.45 | 68.75 ± 2.13 | 70.20 ± 1.43 | 95.39 ± 1.54 | 31.60 ± 1.70 | 34.20 ± 2.30 | 39.30 ± 0.35 |
| SLOG(N) | 76.50 ± 2.60 | 94.40 ± 0.50 | 96.60 ± 0.10 | 84.40 ± 0.80 | 81.00 ± 0.60 | — | 71.90 ± 0.20 | 97.20 ± 0.10 | 38.20 ± 1.20 | **43.02 ± 2.20** | 45.60 ± 0.60 |
| **HMH (ours)** | 81.50 ± 1.60 | **96.80 ± 0.50** | **98.60 ± 0.10** | 87.40 ± .80 | **84.56 ± 1.90** | **76.10 ± 0.70** | **73.30 ± 0.40** | **98.20 ± 0.40** | **39.50 ± .20** | 41.20 ± 0.20 | **48.64 ± 0.55** |

*Table 2.* Graph-classification accuracy (%) comparison on TU datasets. Mean is ±95% CI and '–' means no result found.

| Method | PROTEINS | NCI1 | NCI109 | MUTAG | D&D | IMDB-M | REDDIT-12K | Mutagenicity |
| --- | --- | --- | --- | --- | --- | --- | --- | --- |
| DiffPool | 68.90 ± 2.95 | 77.73 ± 0.83 | 77.13 ± 1.49 | 79.22 ± 1.02 | 78.61 ± 1.32 | 51.31 ± 0.72 | 44.8 ± 1.5 | 80.78 ± 1.12 |
| EigenPool | 70.84 ± 1.06 | 77.24 ± 0.96 | 75.99 ± 1.42 | – | 78.63 ± 1.36 | 49.81 ± 0.48 | 44.23 ± 1.3 | 80.11 ± 0.73 |
| gPool | 71.71 ± 1.75 | 76.25 ± 1.39 | 76.61 ± 1.39 | 67.85 ± 1.38 | 77.02 ± 1.32 | 48.3 ± 1.6 | 44.35 ± 0.76 | 80.30 ± 1.54 |
| SAGPool(G) | 71.72 ± 2.19 | 77.88 ± 1.59 | 75.74 ± 1.47 | 76.78 ± 2.12 | 78.70 ± 2.29 | 49.47 ± 0.56 | 42.3 ± 1.6 | 79.72 ± 0.79 |
| TopKPool | 70.48 ± 1.01 | 67.61 ± 3.36 | 73.63 ± 0.55 | 77.61 ± 3.36 | 73.63 ± 0.55 | 48.59 ± 0.72 | 45.3 ± 1.43 | 82.45 ± 1.32 |
| SEP (Wu et al., 2022) | 75.22 ± 0.63 | 74.83 ± 1.49 | 76.58 ± 1.04 | 85.83 ± 1.49 | 78.58 ± 1.04 | 50.78 ± 0.75 | 45.3 ± 1.6 | 79.1 ± 1.2 |
| Mincut-Pool (Bianchi et al., 2019) | 76.25 ± 0.8 | **82.38 ± 0.4** | 72.32 ± 0.48 | 89.7 ± 1.1 | 79.2 ± 0.8 | 52.8 ± 0.5 | – | 79.34 ± 1.24 |
| MaxCut-Pool (Abate & Bianchi, 2024) | **77.10 ± 2.5** | 83.20 ± 1.3 | – | 79.2 ± 4.5 | **81.3 ± 2.7** | **54.1 ± 2.4** | 43.46 ± 0.45 | 80.34 ± 1.24 |
| HMH (ours) | 76.8 ± 2.1 | 80.9 ± 2.5 | **80.7 ± 2.0** | **94.50 ± 1.8** | 79.7 ± 1.8 | 52.5 ± 2.2 | **46.1 ± 1.2** | **81.7 ± 1.8** |

## 6.1. Hub Domination, Oversmoothing and Oversquashing Analysis

Since degree-dependent aggregation drives hub domination, a degree-stratified evaluation provides a direct stress test: it isolates performance on low-degree nodes ( weaker aggregation influence) versus high-degree nodes (stronger aggregation influence). We divide the test nodes into three groups based on node degree: "Spokes" (low-degree), "Medium" (mid-degree), and "Hubs" (high-degree), as shown in Table 3. GCN and BernNet are significantly more accurate on *Hubs* than on *Spokes*, suggesting hub-dominated aggregation. This disparity is even more pronounced in large-scale graphs (Penn94, Amazon-Ratings): Spoke accuracy can be up to 19% lower than Hub accuracy. Proposed HMH reduces hub dominance by multiscale Haar filtering, increasing Spokes on Penn94 by $+10.8\%$ and minimizing the degree gap compared to baseline. The pattern continues on highly heterogeneous graphs, such as Squirrel (non-filtered), where HMH improves all cohorts (e.g., $+9.1\%$ on Spokes) by preventing oversmoothing of conflicting signals. On the other hand, TOLOKERS exhibits *reverse* hub domination (the baselines perform worse on high-degree nodes, likely due to noisy superusers/bots). Instead of blindly boosting the hub accuracy, HMH improves all cohorts by $+1.6\%$ over baseline, signaling strong adaptation to the graph structure. Overall, HMH prevents amplification of hubs while filtering degree-induced structural noise.

**Degree-cohort protocol.** Since degree distributions vary substantially across datasets, we do not use a single absolute degree cutoff . Instead, we define cohorts relative to each dataset's normalized degree distribution. Nodes in the lowest 15 percentile of the normalized degree distribution are treated as *Spokes*, nodes between the 20th and 70th percentiles are treated as *Medium*, and the remaining high-degree nodes are treated as *Hubs*.

We empirically show that the proposed HMH mitigates two depth-related GNN pathologies: **oversmoothing** and **oversquashing**.

**Oversmoothing** results are in Fig. 4 (a–c). We can see that as depth grows, standard GCNs ' accuracy rapidly degrades (below 20% on PENN94 and ARXIV at 64 layers), consistent with feature homogenization. Even, depth-stabilized baselines (GCNII, EvenNet) still drop by $5–10\%$. In contrast, HMH remains stable and improves with depth, peaking on PENN94 at 32–64 layers (up to $85.1\%$), indicating better preservation of discriminative signals.

For **oversquashing**, we use TREE-NEIGHBORSMATCH (Fig. 4 (d)) benchmark (Pei et al., 2024), where correct leaf predictions require information from distant ancestors. As tree depth increases, the required information must travel farther, creating a strong bottleneck in the information flow from distant neighbors. InFig. 4 (d), we can see baseline models such as GIN and ChebNet quickly lose long-range context and drop to random guess accuracy (50%) by depth 8. In contrast, HMH maintains reliable long-range propagation and strong accuracy ($> 98\%$) even at depth 10. This signals HMH's effective long-range information-propagation

*Table 3.* Stratified Accuracy (%) Analysis on Hub Domination. We report performance across node degree ($d$) cohorts: **Spokes** (Low), **Medium**, and **Hubs** (High). The number of nodes ($N$) and degree range is indicated in the header.

| Model | Penn94 (Social) | | | Squirrel (non filtered Wiki) | | | Tolokers (Crowdsource) | | | Amazon-Ratings (Co-purchase) | | |
|---|---|---|---|---|---|---|---|---|---|---|---|---|
| | ($d \leq 20$ | $20 < d \leq 150$ | $d > 150$) | ($d \leq 15$ | $15 < d \leq 100$ | $d > 100$) | ($d \leq 38$ | $38 < d \leq 470$ | $d > 470$) | ($d \leq 10$ | $10 < d \leq 24$ | $d > 24$) |
| | Spokes ($N = 1197$) | Med. ($N = 5309$) | Hubs ($N = 3199$) | Spokes ($N = 178$) | Med. ($N = 630$) | Hubs ($N = 233$) | Spokes ($N = 1170$) | Med. ($N = 1488$) | Hubs ($N = 282$) | Spokes ($N = 3252$) | Med. ($N = 2264$) | Hubs ($N = 607$) |
| GCN | 71.35 | 82.58 | 90.43 | 34.27 | 31.59 | 37.34 | 81.28 | 78.36 | 70.57 | 42.96 | 40.90 | 50.08 |
| GPRGNN | 73.50 | 81.10 | 88.20 | 36.40 | 33.10 | 42.50 | 80.94 | 78.36 | 69.50 | 44.40 | 41.78 | 49.42 |
| BernNet | 75.80 | 83.20 | 89.10 | 40.50 | 37.20 | 44.80 | 81.50 | 78.90 | 70.10 | 46.20 | 43.50 | 50.80 |
| ChebNetII | 76.40 | 83.90 | 89.50 | 41.20 | 38.50 | 45.40 | 81.80 | 79.10 | 70.80 | 47.10 | 44.80 | 51.20 |
| HMH (Ours) | **82.15** | **88.40** | **91.10** | **52.40** | **48.60** | **51.20** | **83.50** | **80.10** | **72.20** | **50.10** | **47.50** | **52.30** |

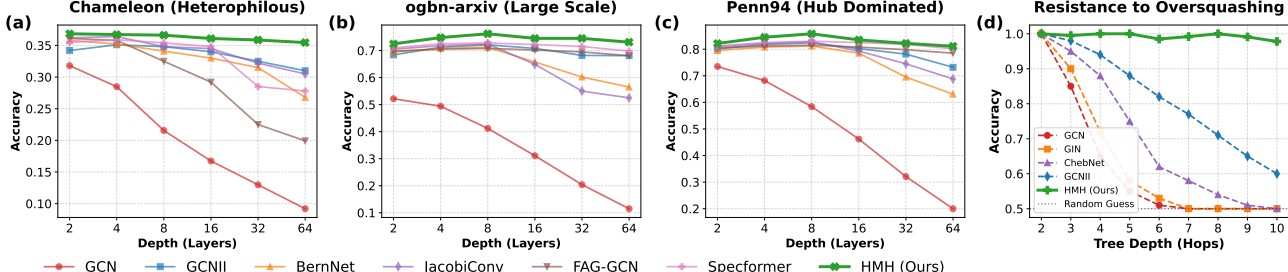

*Figure 4.* **Oversmoothing (a, b, and c) and Oversquashing (d) Analysis.**

capability. Dirichlet energy results are provided in Appendix I. we further evaluate HMH on standard long-range graph benchmarks in Section 6.2.

## 6.2. Long-Range Graph Benchmarks

To examine whether hierarchical coarsening causes information loss on long-range tasks, we evaluate HMH on Peptides-func and Peptides-struct from the Long Range Graph Benchmark. HMH performs competitively with Transformer-based models and improves over standard message-passing GNNs, suggesting that the hierarchy preserves useful long-range information rather than compressing it away.

*Table 4.* Performance on long-range graph benchmarks. Higher AP is better for Peptides-func, and lower MAE is better for Peptides-struct.

| Model | Peptides-func AP ↑ | Peptides-struct MAE ↓ |
|---|---|---|
| GCN | $0.5930 \pm 0.0023$ | $0.3496 \pm 0.0013$ |
| GCNII | $0.5543 \pm 0.0078$ | $0.3471 \pm 0.0010$ |
| GINE | $0.5498 \pm 0.0079$ | $0.3547 \pm 0.0045$ |
| GatedGCN | $0.5864 \pm 0.0077$ | $0.3420 \pm 0.0013$ |
| Transformer+LapPE | $0.6326 \pm 0.0126$ | $0.2529 \pm 0.0016$ |
| HMH (ours) | $0.6220 \pm 0.0115$ | $0.2350 \pm 0.0023$ |

## 6.3. Graph Classification via Spectral Pooling

HMH is ideal for graph classification because its hierarchical coarsening explicitly implements pooling: at each level, nodes are aggregated into supernodes, and the final (coarsest) level provides a compact graph-level embedding. **Experimental Setup.** We evaluate HMH on 7 benchmarks covering molecular (e.g., MUTAG, NCI1) and social networks (IMDB, REDDIT). We compare against strong

pooling baselines DiffPool (Ying et al., 2018), SAGPool (Lee et al., 2019), TopKPool (Diehl, 2019) and graph classifiers DGCNN (Phan et al., 2018), SEP (Wu et al., 2022). Hyperparameters, including loss weights ($\lambda_{\text{div}}$, $\lambda_{\text{rec}}$) and coarsening depth $\ell$ are detailed in Appendix G.

**Graph Classification Results.** Table 2 reports graph classification accuracy across bioinformatics and social benchmarks. HMH achieves strong gains on structure-heavy bioinformatics graphs, reaching **94.5%** on MUTAG and consistently improving on PROTEINS and NCI1, suggesting that high-frequency Haar components capture fine-grained molecular signals that are often over-smoothed by purely low-pass pooling. On hub-heavy social graphs (e.g., IMDB-M, REDDIT-12K), HMH remains highly competitive, indicating that the learned orthogonal basis can disentangle community structure while avoiding signal mixing. Full experimental settings and dataset details are provided in the Appendix G.

## 7. Ablation Study and Runtime Analysis

We ablate three components of HMH under the same protocol as the full model. 1) **Fixed adjacency** replaces the adaptive signed affinity with the original $A$. 2) **No hierarchy** performs Haar filtering only at $\ell=0$ (no coarsening / no multiscale fusion). 3) **Fixed basis** replaces the learned localized Haar basis with a Chebyshev global surrogate. Table 5 shows consistent drops, with the largest degradations on heterophilous graphs (Actor/Chameleon/Squirrel), supporting the roles of adaptive signing, hierarchy, and localized orthonormal bases. Additional ablations are in

Appendix J. Moreover, HMH runs in near-linear time on sparse graphs since both $A_{\mathrm{adp}}$ and we report wall-clock time and peak memory on REDDIT in Appendix H.

*Table 5.* Ablation on HMH (mean±std, %). We used the non-filtered version of the squirrel and chameleon datasets for ablation.

| Variant | PubMed | Actor | Chameleon | Squirrel | Flickr | OGBN |
|---|---|---|---|---|---|---|
| HMH (full) | 91.4±0.5 | 43.3±0.7 | 41.8±1.2 | 50.9±1.4 | 51.3±0.2 | 73.3±0.4 |
| Fixed adjacency | 87.3±0.6 | 39.8±1.3 | 37.0±1.8 | 47.0±1.5 | 49.6±0.3 | 71.9±0.5 |
| No hierarchy | 89.8±0.6 | 37.5±1.3 | 34.2±1.7 | 45.4±1.4 | 50.1±0.3 | 72.4±0.4 |
| Fixed basis | 88.9±0.6 | 38.1±1.3 | 37.2±1.7 | 48.3±1.5 | 50.8±0.2 | 72.8±0.4 |

### 7.1. Sensitivity Analysis

The coarsening ratio $R$ controls the number of nodes retained at each hierarchical level. Larger values preserve more fine-scale nodes, while smaller values produce more aggressive coarsening. Table 6 reports performance for different values of $R$. Overall, HMH performs best with moderate coarsening, especially $R \in [0.4, 0.5]$. Large values of coarsening ratio $R$ reduce the benefit of multi-scale interaction, while very small values may compress local information too aggressively.

*Table 6.* Sensitivity analysis for the coarsening ratio $R$.

| Dataset | $R = 0.8$ | $R = 0.6$ | $R = 0.5$ | $R = 0.4$ | $R = 0.3$ |
|---|---|---|---|---|---|
| Physics | 94.9 | 98.2 | 98.6 | 98.5 | 98.1 |
| Roman-empire | 72.6 | 75.3 | 75.8 | 76.1 | 75.4 |
| Squirrel | 34.8 | 38.6 | 39.1 | 39.5 | 38.7 |
| Ogbn-arxiv | 70.1 | 72.8 | 73.3 | 73.0 | 72.4 |
| PROTEINS | 73.2 | 76.0 | 76.8 | 76.5 | 75.7 |
| NCI109 | 74.1 | 79.9 | 80.3 | 80.7 | 80.0 |
| MUTAG | 88.8 | 93.6 | 94.5 | 94.1 | 93.2 |
| REDDIT-12K | 41.8 | 45.4 | 45.8 | 46.1 | 45.2 |

Moreover, we have presented stability of the algorithm under noisy features in Appendix J.1

## 8. Limitations

Our current formulation assumes a static graph. Extending HMH to dynamic settings with frequently changing edges would require updating the hierarchy over time. Also, extending HMH to multi-relational/directed graphs is an important direction for broader graph learning.

## 9. Conclusion

We presented HMH, a sign-aware, multi-resolution framework designed for heterophilous graphs. Using an adaptive encoder that integrated structural and feature affinities into a unified node score, HMH enabled systematic hierarchical coarsening and constructed sparse, orthonormal Haar bases at each level in almost linear time. These locality-preserving orthogonal bases confine energy strictly

to two-hop neighborhoods, allowing the framework to distinguish low-frequency (homophilous) from high-frequency (heterophilous) signals across clusters of varying sizes without relying on costly eigen-decomposition. Through this design, HMH mitigated hub-aliasing and basis-suboptimality, achieved state-of-the-art performance on diverse node- and graph-classification benchmarks, and significantly reduced preprocessing overhead. Analytical results, rigorous proofs, and experimental validation confirmed the proposed approach, showing that HMH consistently outperformed state-of-the-art spectral baselines – achieving up to a 3% improvement in node-classification accuracy on heterophilous datasets and a 7% gain on graph-classification tasks, while maintaining linear scalability.

## Impact Statement

This work contributes to graph representation learning for graphs that are heterophilous, degree-imbalanced, or structurally diverse. HMH is designed to reduce oversmoothing, oversquashing, and hub domination, which can help preserve useful information from low-degree or structurally underrepresented regions of a graph. This may be useful in applications such as recommender systems, biological networks, molecular graphs, citation networks, and social network analysis. Moreover, HMH is intended as a methodological contribution for scalable and robust graph learning. It should not be used as a standalone decision-making system in high-stakes settings without domain-specific validation, fairness checks, uncertainty analysis, and human oversight.

## Acknowledgments

The research is partially supported by National Science Foundation under grant Nos. # 2327409 and #2337999, and the Alabama Graduate Research Scholars Program (GRSP), funded through the Alabama Commission for Higher Education and administered by the AL EPSCoR.

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

# A. Proof of Theorem 4.1

Consider three clusters $A$, $B$, and $H'$. The tiny clusters or spokes $A$ and $B$ of size $a = |A|$ and $b = |B|$) are attached to a common hub $H'$ of size $M = |H'|$ with $M \gg a, b$. Let the cluster mean features given by $\mu_A \in \mathbb{R}^d$, $\mu_B \in \mathbb{R}^d$, $\mu_{H'} \in \mathbb{R}^d$, respectively for $A$, $B$, and $H'$. Define $\delta = \|\mu_A - \mu_B\|$ that measures the separation $A$–$B$ and $\Delta = \left\|\mu_H - \frac{\mu_A + \mu_B}{2}\right\|$ that measures the hub offset relative to the midpoint of $A$ and $B$

To analyze hub aliasing coherently, we will use the following standard assumptions:

**Assumption 1** (i) The hub's mean feature is *significantly different* from spokes $A$ and $B$ features, i.e., $\Delta \geq \kappa\delta$ for some margin $\kappa > 1$.

(ii) The hub's size $M$ is sufficiently larger than the combined size with the weighed geometric factor, i.e., $\Delta/\delta$,i.e., $M \geq \frac{(a+b)\Delta}{\delta}(1 + \varepsilon)$ for some $\varepsilon > 0$.

We will consider the CMA and SMP algorithms in two cases to demonstrate that hub aliasing occurs in both the cases.

**Case I: Hub-aliasing in CMA:** For the *CMA layer mechanics*, the projection of the initial embedding is given by

$$z_i = H_i^{(\ell-1)}W \in \mathbb{R}^{d'}.$$

The pre-softmax scores (per edge and per chunk) is given by

$$g(z_i, z_j) = \tau^{-1} relu(\alpha z_i + z_j) W_{\text{att}} \in \mathbb{R}^C. \tag{11}$$

The soft labels (chunk weights) is given by

$$s_{ij} = \text{softmax}\big(g(z_i, z_j)\big) \in \Delta^{C-1}. \tag{12}$$

The chunked aggregation is given by $C_{i,t} = \sum_{j \in \mathcal{N}(i)} s_{ij,t} z_j$, and

$$m_i = \big\|_{t=1}^{C} C_{i,t}. \tag{13}$$

The CMA update rule is expressed by

$$H_i^{(\ell)} = relu\big((1 - \beta)H_i^{(0)} + \beta\, m_i\big) \tag{14}$$

The projection embedding is $z_h \approx \mu_H W$ for $h \in H'$, $z_u \approx \mu_A W$ for $u \in A$, $z_v \approx \mu_B W$ for $v \in B$.

Now using Lipschitz and strong monotonicity, we get the following: ReLU is 1-Lipschitz. So,the map $x \mapsto xW_{\text{att}}$ is $\|W_{\text{att}}\|_2$-Lipschitz. Scaling by $\tau^{-1}$ is $\tau^{-1}$-Lipschitz. Hence, for the score map $g$ using (11) we get,

$$\|g(\alpha z_i + z_h) - g(\alpha z_j + z_h)\| \leq \frac{\|W_{\text{att}}\|_2}{\tau} \alpha \|W\|_2 \|z_i - z_j\|. \tag{15}$$

Softmax is $L_\sigma$-Lipschitz with $L_\sigma \leq 1$ and on the subspace orthogonal to $\mathbf{1}$ it is $\kappa_\sigma$-strongly monotone, given by

$$\|\text{softmax}(u) - \text{softmax}(v)\| \geq \kappa_\sigma \|u - v\|. \tag{16}$$

where $\kappa_\sigma := p_{\min}(1 - p_{\min})$ with $p_{\min} > 0$ is the minimum probability of softmax output, and we assume no collapse of $W$ along $(\mu_A - \mu_H)$.

For spoke to hub edges $(u, h)$ versus $(v, h)$, we can calculate the difference of the soft label using (12) as

$$\|s_{uh} - s_{vh}\| \leq L_\sigma \frac{\|W_{\text{att}}\|_2}{\tau} \alpha \|W\|_2 \|z_u - z_v\| \leq c_{\text{att}}\, \delta. \tag{17}$$

Similarly, for spoke versus hub edges $(u, w)$ and $(h, w)$ with $w \in H'$, we can calculate the difference of the soft label using (12) as

$$\|s_{uw} - s_{hw}\| \geq \kappa_\sigma \frac{\|W_{\text{att}}\|_2}{\tau} \alpha \|W\|_2 \|z_u - z_h\| \geq c'_{\text{att}}\, \Delta, \tag{18}$$

where $c_{\text{att}} = L_\sigma \frac{\|W_{\text{att}}\|_2}{\tau} \alpha \|W\|_2$ and $c'_{\text{att}} = \kappa_\sigma \frac{\|W_{\text{att}}\|_2}{\tau} \alpha \|W\|_2$. By partitioning the message update for the node into hub and spoke components as per (13) and aggregating over neighbors, we obtain

$$\|m_u - m_v\| \le M\, c_{\text{att}}\, \delta\, \|z_H\| + (a+b)\, \|W\|_2\, \delta = K_z\, \delta, \tag{19}$$

$$\|m_u - m_h\| \ge M\, c'_{\text{att}}\, \Delta\, \|z_H\| = K_h\, \Delta, \tag{20}$$

where $K_z = M\, c_{\text{att}}\, \|z_H\| + (a+b)\, \|W\|_2$ and $K_h = M\, c'_{\text{att}}\, \|z_H\|$.

Recalling the CMA hub update from (14) we get,

$$\|H_u^{(1)} - H_v^{(1)}\| \le (1-\beta)\, \delta + \beta\, K_z\, \delta, \tag{21}$$

$$\|H_u^{(1)} - H_h^{(1)}\| \ge \beta\, K_h\, \Delta - (1-\beta)\, \Delta. \tag{22}$$

Therefore dividing (21) by (22) we get

$$\frac{\|H_u^{(1)} - H_v^{(1)}\|}{\|H_u^{(1)} - H_h^{(1)}\|} \le \frac{(1-\beta) + \beta K_z}{\beta K_h - (1-\beta)}\, \frac{\delta}{\Delta}. \tag{23}$$

Substituting $K_z = M\, c_{\text{att}}\, \|z_H\| + (a+b)\, \|W\|_2$, $K_h = M\, c'_{\text{att}}\, \|z_H\|$, and grouping the hub size terms, it holds that

$$\frac{(1-\beta) + \beta K_z}{\beta K_h - (1-\beta)} \le \frac{1-\beta}{\beta K_h} + \frac{c_{\text{att}}}{c'_{\text{att}}} + \frac{a+b}{M}\, \frac{\|W\|_2}{c'_{\text{att}}\|z_H\|} \tag{24}$$

$$= \theta_0 + \kappa + \rho\, \theta_1, \tag{25}$$

where $\theta_0 = \frac{1-\beta}{\beta K_h}$, $\kappa = \frac{c_{\text{att}}}{c'_{\text{att}}}$ and $\theta_1 = \frac{\|W\|_2}{c'_{\text{att}}\|z_H\|}$. For large $M$, $\theta_0$ is negligible.

Define $\eta := \kappa + \rho\, \theta_1$. From the definition $c_{\text{att}} < c'_{\text{att}}$ and $\rho < 1$, and so $\eta < 1$. Combining (23) and (25) yields

$$\frac{\|H_u^{(1)} - H_v^{(1)}\|}{\|H_u^{(1)} - H_h^{(1)}\|} \le \eta\, \frac{\delta}{\Delta}. \tag{26}$$

Repeating the same argument at each layer for $L$ layers with the same constants or their upper envelopes, we obtain

$$\frac{\|H_u^{(L)} - H_v^{(L)}\|}{\|H_u^{(L)} - H_h^{(L)}\|} \le (\eta\, \frac{\delta}{\Delta})^L, \tag{27}$$

with $0 < \frac{\delta}{\Delta} < 1$. From (27), we can state that the spoke–spoke separation decays exponentially relative to the spoke–hub separation across layers. Therefore, CMA concentrates information toward the hub.

**Case II: Hub-aliasing in SMP:** Let the signed adjacency $S_{uv} = +1$ if $(u,v) \in E$ and $y_u = y_v$, and $S_{uv} = -1$ if $y_u \ne y_v$. The SMP update (omitting bias and ReLU for upper bound) is given by

$$H^{(k+1)} = S H^{(k)} W^{(k)}, \quad \|W^{(k)}\|_2 = 1, \quad H^{(0)} = X. \tag{28}$$

So we can express

$$\left\| S\, H^{(k)} W^{(k)} \right\|_2 \le \left\| S\, H^{(k)} \right\|_2.$$

In addition, two consecutive steps of the SMP can be expressed as

$$H^{(k+2)} = S\big(S H^{(k)} W^{(k)}\big) W^{(k+1)} \approx S^2\, H^{(k)}.$$

Now, there are no edges between $A$ and $B$S, so a walk from a node in $A$ to another region must pass through $H'$. Similarly, a walk from a node in $B$ to another region must pass through $H'$. Hence for $u \in A$, $v \in B$, and any $h \in H'$ the second layer embedding can be given by

$$h_u^{(2)} = (S^2 X)_u = \sum_{w \in H} \sum_{z \in N(w) \cap A} X_z$$
$$+ \sum_{z \in N(u) \cap A} \sum_{w \in N(z) \cap H} X_w \approx M \mu_H + (a-1)\, \mu_A, \tag{29}$$

Similarly, for node $v$ and $h$ we have

$$h_v^{(2)} \approx M \, \mu_H' + (b-1) \, \mu_B, \tag{30}$$

$$h_h^{(2)} \approx M \, \mu_H' + (a-1) \, \mu_A + (b-1) \, \mu_B. \tag{31}$$

By subtracting and calculating the norm of (29) and (30) we get

$$\begin{aligned} \|h_u^{(2)} - h_v^{(2)}\|_2 &\le \|(a-1)\,\mu_A - (b-1)\,\mu_B\|_2 \\ &\le (a+b) \, \|\mu_A - \mu_B\|_2 =: (a+b)\,\delta. \end{aligned} \tag{32}$$

By subtracting and calculating the norm of (29) and (31) we have

$$\|h_u^{(2)} - h_h^{(2)}\|_2 \ge \left\| M\,\mu_H - \big((a-1)\mu_A + (b-1)\mu_B\big) \right\|_2 - (a+b)\,\delta \ge M\,\Delta - (a+b)\,\delta, \tag{33}$$

Dividing (32) by (33) gives

$$\frac{\|h_u^{(2)} - h_v^{(2)}\|_2}{\|h_u^{(2)} - h_h^{(2)}\|_2} \le \frac{(a+b)\,\delta}{M\,\Delta - (a+b)\,\delta} \le \frac{\rho}{1+\epsilon}, \quad \rho = \frac{a+b}{M}. \tag{34}$$

So that after $L = 2k$ layers (34) becomes,

$$\frac{\|h_u^{(L)} - h_v^{(L)}\|_2}{\|h_u^{(L)} - h_h^{(L)}\|_2} \le \left(\frac{\rho}{1+\epsilon}\right)^k = \left(\frac{\rho}{1+\epsilon}\right)^{\lceil L/2 \rceil}, \tag{35}$$

where $\frac{\rho}{1+\epsilon} < 1$. Therefore, spoke-spoke separation diminishes exponentially in relation to the spoke-hub separation across the layer of SMP. This completes the proof. ∎

## B.1 Proof of Theorem 5.1

Signed Message Passing (SMP) with a fixed signed adjacency matrix $S$ calculates the embedding of each layer as

$$H^{(\ell+1)} = S H^{(\ell)}.$$

Then for the $\ell + 2$ layer, we can write

$$H^{(\ell+2)} = S^2 H^{(\ell)} = H^{(\ell)}.$$

In particular, after two-hop propagation, the sign pattern may become fully positive when

$$S^2 = c\mathbf{1}_{n \times n},$$

where $c$ is a constant determined by the magnitude of the similarity weights. For example, let $n = 2$ and $S = \begin{bmatrix} -1 & -1 \\ -1 & -1 \end{bmatrix}$. Then

$$S^2 = \begin{bmatrix} 2 & 2 \\ 2 & 2 \end{bmatrix} = 2\mathbf{1}_{2 \times 2}.$$

Thus, the first propagation through negative edges produces a negative signed aggregation, while the second propagation produces a fully positive aggregation. This illustrates the two-step sign-flipping effect in fixed signed message passing.

In the HMH encoder, the signed adaptive adjacency is recomputed at each layer as

$$A_{\text{adp}}^{(\ell)} = \left(2\widetilde{S}^{(\ell)} - 1\right) \odot A,$$

where

$$\widetilde{S}_{ij}^{(\ell)} = \text{softmax}_{j \in \mathcal{N}(i)} \left[ S_{\text{att}}^{(\ell)}(i,j) + S_{\text{struct}}^{(\ell)}(i,j) \right].$$

Therefore, $A_{\mathrm{adp}}^{(\ell)}$ depends on the current representation $H^{(\ell)}$ and, in general,

$$A_{\mathrm{adp}}^{(\ell+1)} \neq A_{\mathrm{adp}}^{(\ell)}.$$

The layer-wise update is

$$H^{(\ell+1)} = A_{\mathrm{adp}}^{(\ell)} H^{(\ell)}.$$

After two layers, we have

$$H^{(\ell+2)} = A_{\mathrm{adp}}^{(\ell+1)} A_{\mathrm{adp}}^{(\ell)} H^{(\ell)}.$$

In HMH, the two-step operator is not the square of a fixed signed matrix. Instead, it is a product of two layer-dependent adaptive matrices. For a fixed two-step sign flip or fixed positive recovery pattern to persist, one would need a layer-independent relationship such as

$$A_{\mathrm{adp}}^{(\ell+1)} A_{\mathrm{adp}}^{(\ell)} = c\mathbf{1}_{n \times n}$$

or

$$A_{\mathrm{adp}}^{(\ell+1)} A_{\mathrm{adp}}^{(\ell)} = \sigma I, \qquad \sigma \in \{-1, 1\},$$

uniformly across layers. However, this condition is not imposed in HMH because each $A_{\mathrm{adp}}^{(\ell)}$ is recomputed from the current embeddings and local structural similarities. Hence, the model is not forced to follow the repeated power pattern of a fixed signed matrix. Therefore, HMH avoids the rigid two-step sign oscillation of fixed SMP by replacing fixed signed propagation with layer-wise adaptive signed propagation. This completes the proof.

## B. Proof of Theorem 5.2

We analyze the eigenpairs of the original (pre-encoder) Laplacian $L$, denoted as $\{(\lambda_k, u_k)\}_{k=1}^n$ with $\lambda_1 \leq \cdots \leq \lambda_n$. To examine the spectral impact of the adaptive Laplacian, we divide the spectrum at index $\tau$ into the low-frequency subspace $U_{\mathrm{low}} = \mathrm{span}\{u_1, \ldots, u_\tau\}$ and its orthogonal complement $U_{\mathrm{high}} = \mathrm{span}\{u_{\tau+1}, \ldots, u_n\}$. We partition the edge set into $E_{\mathrm{same}} = \{(i,j) \in E : y_i = y_j\}$ homophilous edges and $E_{\mathrm{diff}} = E \setminus E_{\mathrm{same}}$ hetereophilous edges. We now define the bounds for the similarity score as

$$\begin{aligned}
\epsilon_\ell &= 1 - \min_{(i,j) \in E_{\mathrm{same}}} S_{ij}^{(\ell)}, \\
\alpha_\ell &= 1 - \max_{(i,j) \in E_{\mathrm{diff}}} S_{ij}^{(\ell)},
\end{aligned} \tag{36}$$

where $\epsilon_\ell$ indicates how much we can lower the weight of the "most similar" same-label edge and $\alpha_\ell$ tells us how much the "least similar" edge with a different label can still drive the signal toward homophily. For layer $\ell$, let $S_{ij}^{(\ell)} \in [0, 1]$ be the encoder similarity and define the adaptive Laplacian

$$L_{\mathrm{adp}}^{(\ell)} := L + \Delta^{(\ell)},$$

where the perturbation is specified *entrywise* by

$$\Delta_{ij}^{(\ell)} = \begin{cases} -\left(1 - S_{ij}^{(\ell)}\right), & (i,j) \in E_{\mathrm{same}}, \\ \left(1 - S_{ij}^{(\ell)}\right), & (i,j) \in E_{\mathrm{diff}}, \\ 0, & i = j \text{ or } (i,j) \notin E. \end{cases} \tag{37}$$

For any $(i,j) \in E_{\mathrm{same}}$, by definition $S_{ij}^{(\ell)} \geq \min_{E_{\mathrm{same}}} S_{pq}^{(\ell)}$, which implies $1 - S_{ij}^{(\ell)} \leq 1 - \min_{E_{\mathrm{same}}} S_{pq}^{(\ell)} = \varepsilon_\ell$.

For the case $\Delta_{ij}^{(\ell)} = -(1 - S_{ij}^{(\ell)}) \; \forall (i,j) \in E_{\mathrm{same}}$ according to (37), it follows that $-\varepsilon_\ell \leq \Delta_{ij}^{(\ell)} \leq 0$. Again for $(i,j) \in E_{\mathrm{diff}}$, by definition $S_{ij}^{(\ell)} \leq \max_{E_{\mathrm{diff}}} S_{pq}^{(\ell)}$. So $1 - S_{ij}^{(\ell)} \geq 1 - \max_{E_{\mathrm{diff}}} S_{pq}^{(\ell)} = \alpha_\ell$ and also $1 - S_{ij}^{(\ell)} \leq \varepsilon_\ell$.

For the case $\Delta_{ij}^{(\ell)} = +(1 - S_{ij}^{(\ell)}) \; \forall (i,j) \in E_{\mathrm{diff}}$ according to (37), we obtain

$$\alpha_\ell \leq \Delta_{ij}^{(\ell)} \leq \varepsilon_\ell. \tag{38}$$

Lets denote $\{\mu_k^{(\ell)}\}_{k=1}^n$ be the eigenvalues of $L_{\text{adp}}^{(\ell)} = L + \Delta^{(\ell)}$. To prove that Adaptive Adjacency works as high high-pass and low-pass filter, we will show the following:

(i) Low-frequency (smooth) modes shift as $\mu_k^{(\ell)} \le \lambda_k + \varepsilon_\ell \lambda_\tau$   for $k \le \tau$. So the low-pass band remains essentially intact.

(ii) High-frequency modes receive a uniform positive lift as, $\mu_k^{(\ell)} \ge \lambda_k + \beta_\ell \lambda_{\tau+1}$, , where $\beta > 0$, thereby sharpening high-pass separation and enhancing heterophilous contrast.

According to the graph spectral theory (Chung, 1997), for any $x \in \mathbb{R}^n$, we expand

$$x^\top \Delta^{(\ell)} x = \tfrac{1}{2} \sum_{(i,j) \in E} \Delta_{ij}^{(\ell)} (x_i - x_j)^2. \tag{39}$$

According to the (37) the edge-wise bound $\Delta_{ij}^{(\ell)} \ge -\varepsilon_\ell$ on $E_{\text{same}}$ and $\Delta_{ij}^{(\ell)} \le \varepsilon_\ell$ on $E_{\text{diff}}$, it follows that

$$\begin{aligned} x^\top \Delta^{(\ell)} x \le\ & -\tfrac{\varepsilon_\ell}{2} \sum_{(i,j) \in E_{\text{same}}} (x_i - x_j)^2 \\ & + \tfrac{\varepsilon_\ell}{2} \sum_{(i,j) \in E_{\text{diff}}} (x_i - x_j)^2. \end{aligned} \tag{40}$$

For any $x \in \mathbb{R}^n$ and the total energy is $Q = Q_{\text{same}} + Q_{\text{diff}}$ (Chung, 1997), we can define the edge energy splits as

$$Q_{\text{same}}(x) = \sum_{(i,j) \in E_{\text{same}}} (x_i - x_j)^2, \tag{41}$$

$$Q_{\text{diff}}(x) = \sum_{(i,j) \in E_{\text{diff}}} (x_i - x_j)^2. \tag{42}$$

So using (39) and substituting (42), the total energy is

$$Q(x) = Q_{\text{same}}(x) + Q_{\text{diff}}(x) = 2x^\top L x. \tag{43}$$

Let $x \in U_{\text{low}} \setminus \{0\}$. By the Rayleigh quotient (Chung, 1997),

$$\frac{x^\top L x}{\|x\|^2} \le \lambda_\tau. \tag{44}$$

Substituting the value of $Q(x)$ from (43) into (44) we get, (44) implies

$$Q(x) \le 2\lambda_\tau \|x\|^2. \tag{45}$$

Let define $\rho := |E_{\text{same}}|/|E| \in (0,1)$ the homophily ratio (for a heterophilous graph, $0 < \rho < \tfrac{1}{2}$). By edge counting theory (Chung, 1997),

$$\frac{Q_{\text{diff}}(x)}{Q(x)} \le 1 - \rho \implies Q_{\text{diff}}(x) \le (1-\rho) Q(x). \tag{46}$$

Combining (46) with (45) yields

$$Q_{\text{diff}}(x) \le 2(1-\rho)\lambda_\tau \|x\|^2, \tag{47}$$

i.e.,

$$\sum_{(i,j) \in E_{\text{diff}}} (x_i - x_j)^2 \le 2(1-\rho)\lambda_\tau \|x\|^2. \tag{48}$$

Similarly, from $Q_{\text{same}}(x) = Q(x) - Q_{\text{diff}}(x)$ and (46),

$$Q_{\text{same}}(x) \ge \rho Q(x) \overset{(45)}{\ge} 2\rho\lambda_\tau \|x\|^2, \tag{49}$$

i.e.,

$$\sum_{(i,j)\in E_{\text{same}}} (x_i - x_j)^2 \geq 2\rho\,\lambda_\tau\,\|x\|^2. \tag{50}$$

Substituting the bounds (50) and (48) into the quadratic form bound (40), we have

$$\begin{aligned}
x^\top \Delta^{(\ell)} x &\leq -\tfrac{\varepsilon_\ell}{2} \cdot 2\rho\lambda_\tau\|x\|^2 + \tfrac{\varepsilon_\ell}{2} \cdot 2(1-\rho)\lambda_\tau\|x\|^2 \\
&= \varepsilon_\ell\big(1-\rho-\rho\big)\lambda_\tau\|x\|^2 \\
&= \varepsilon_\ell(1-2\rho)\lambda_\tau\|x\|^2.
\end{aligned} \tag{51}$$

Alternatively, we have

$$\frac{x^\top \Delta^{(\ell)} x}{\|x\|^2} \leq \varepsilon_\ell\lambda_\tau, \tag{52}$$

Applying the Courant-Fischer min-max principle (Chung, 1997) for $L_{adp}^\ell$, for any eigen value $\mu_k$ we get,

$$\mu_k^{(\ell)} = \min_{\dim S = k} \max_{0 \neq x \in S} \frac{x^\top (L + \Delta^{(\ell)})x}{\|x\|^2}. \tag{53}$$

**Case I: Low pass filter:** Let $S = U_{\text{low}} = \text{span}\{u_1, \ldots, u_\tau\}$, so $\dim S \geq k$. Applying (53) to $L_{adp}$ for any $x \in U_{\text{low}} \setminus \{0\}$ and substituting from (52) we get,

$$\frac{x^\top (L + \Delta^{(\ell)})x}{\|x\|^2} \leq \frac{x^\top L x}{\|x\|^2} + \frac{x^\top \Delta^{(\ell)} x}{\|x\|^2} \leq \lambda_k + \varepsilon_\ell\lambda_\tau.$$

Taking the minimum over all such $S$, according to (53) yields,

$$\mu_k^{(\ell)} \leq \lambda_k + \varepsilon_\ell\lambda_\tau, \qquad k \leq \tau. \tag{54}$$

**Case II: High pass filter:** According to (37) for any $x \in \mathbb{R}^n$ we can express

$$x^\top \Delta^{(\ell)} x \geq -\frac{\varepsilon_\ell}{2} Q_{\text{same}}(x) + \frac{\alpha_\ell}{2} Q_{\text{diff}}(x). \tag{55}$$

Using $Q_{\text{diff}}(x) = Q(x) - Q_{\text{same}}(x)$ in (55) gives

$$\begin{aligned}
x^\top \Delta^{(\ell)} x &\geq -\frac{\varepsilon_\ell}{2} Q_{\text{same}}(x) + \frac{\alpha_\ell}{2}\big[Q(x) - Q_{\text{same}}(x)\big] \\
&= \frac{1}{2}\Big[\alpha_\ell\, Q(x) - \big(\alpha_\ell + \varepsilon_\ell\big) Q_{\text{same}}(x)\Big] \tag{56} \\
&\geq \frac{1}{2}\Big[\alpha_\ell(1-\rho) - \varepsilon_\ell\rho\Big] Q(x) \tag{57} \\
&\geq \big[\alpha_\ell(1-\rho) - \varepsilon_\ell\rho\big] \lambda_{\tau+1}\|x\|^2. \tag{58}
\end{aligned}$$

Applying (53) to $L_{adp}$ for any $x \in U_{\text{low}} \setminus \{0\}$ and substituting from (58) we get,

$$\begin{aligned}
x^\top\big(L + \Delta^{(\ell)}\big)x &\geq \lambda_{\tau+1}\|x\|^2 + \big[\alpha_\ell(1-\rho) - \varepsilon_\ell\rho\big] \\
&\qquad\qquad\qquad \lambda_{\tau+1}\|x\|^2 \\
&= \big[1 + \alpha_\ell(1-\rho) - \varepsilon_\ell\rho\big] \lambda_{\tau+1}\|x\|^2. \tag{59}
\end{aligned}$$

Therefore, from (59) we have

$$\frac{x^\top (L + \Delta^{(\ell)})x}{\|x\|^2} \geq \frac{x^\top L x}{\|x\|^2} + \frac{x^\top \Delta^{(\ell)} x}{\|x\|^2} \geq \lambda_k + \alpha_\ell(1-\rho)\lambda_{\tau+1}.$$

Applying the Courant-Fischer theorem for $k > \tau$ we get, $\mu_k^{(\ell)} \geq \lambda_k + \alpha_\ell(1 - \rho)\lambda_{\tau+1}$, $\qquad k > \tau$.

Thus, for both cases,

$$\begin{cases} \mu_k^{(\ell)} \leq \lambda_k + \varepsilon_\ell \lambda_\tau, & k \leq \tau, \\ \mu_k^{(\ell)} \geq \lambda_k + \alpha_\ell(1 - \rho)\lambda_{\tau+1}, & k > \tau. \end{cases} \tag{60}$$

Therefore, each low pass frequency is changed only very smaller amount $\varepsilon_\ell << 1$ and each high-frequency eigenvalue of $L_{ada}(k > \tau)$ is *raised* by at least $\left[\alpha_\ell(1 - \rho) - \varepsilon_\ell \rho\right]\lambda_{\tau+1} > 0$. This means that the adaptive Laplacian makes those modes harder to smooth away and boosts the eigenvalue associated with high frequency mode. This completes the proof. ∎

## C. Proof of Theorem 5.3

Consider two disjoint small clusters $A, B \subset V$, and a large hub region $H' \subset V$, such that $A \cap B = \varnothing$, $\quad A \cap H' = \varnothing$, $\quad B \cap H' = \varnothing$. Let $x_A, x_B \in \mathbb{R}^{|V|}$ be denoted as unit-norm normalized indicator signal vectors that are supported on clusters $A$ and $B$. It means that $(x_A)_i = 0$ for every node $i \notin A$.

Now consider a single linear filter defined as $\mathcal{F} : \mathbb{R}^{|V|} \to \mathbb{R}^{|V|}$. A stack of $L$ layers of Filter is defined as $\mathcal{F}^{(L)} = \mathcal{F}_L \cdots \mathcal{F}_1$, where each $\mathcal{F}_\ell$ is a linear filter. We define the separation ratio of two cluster signals after $L$ filtering layers to pre filter embeddings as

$$r(L) = \frac{\left\| \mathcal{F}^{(L)} x_A - \mathcal{F}^{(L)} x_B \right\|_2}{\|x_A - x_B\|_2},$$

where $\mathcal{F}^{(L)} x_A$ is the filter acting on region A and $\mathcal{F}^{(L)} x_B$ acting on region B. The filter $F^{(L)}$ is diagonal in the localized orthonormal basis, so it rescales each coordinate by gains $g_k$. We will show that $\langle F^{(L)} x_A, F^{(L)} x_B \rangle = 0$, i.e., no cross-region leakage. We will also demonstrate that the separation between the two signals after filtering is preserved up to a scaling factor, i.e., $g_{\min} \leq r(L) \leq g_{\max}$. This means filter restricts regional patterns and preserves their contrast, altering only their magnitude by diagonal gains.

Let $B' = [b_1', \ldots, b_n'] \in \mathbb{R}^{n \times n}$ be an *orthogonal, spatially localized* basis ($B'^\top B' = B'B'^\top = I$) such that each column $b_k'$ is supported entirely in exactly one of $A$, $B$, or $H$. Define disjoint index sets as $\Omega_A := \{k : \operatorname{supp}(b_k') \subseteq A\}$, $\quad \Omega_B := \{k : \operatorname{supp}(b_k') \subseteq B\}$, $\quad \Omega_H := \{1, \ldots, n\} \setminus (\Omega_A \cup \Omega_B)$. Assume each layer $F_\ell$ is diagonal in $B'$:

$$F_\ell = B' \operatorname{diag}(h^{(\ell)}) B'^\top, \qquad \ell = 1, \ldots, L.$$

Hence the stack

$$F^{(L)} := F_L \cdots F_1 = B' \operatorname{diag}(g) B'^\top, \qquad g_k := \prod_{\ell=1}^{L} h_k^{(\ell)}.$$

Any signal $X \in \mathbb{R}^{|V|}$ can be represented as $X = B'c$, with the coefficients defined by $c = B'^\top x$. Due to localization we can write,

$$\begin{aligned} x_A = B'c_A & \quad \text{where} \quad (c_A)_k = 0 \ \ \forall k \notin \Omega_A, \\ x_B = B'c_B & \quad \text{where} \quad (c_B)_k = 0 \ \ \forall k \notin \Omega_B. \end{aligned}$$

Moreover, the signals are orthogonal, resulting in the inner product $\langle x_A, x_B \rangle = c_A^\top c_B = 0$ (orthonormality). Then the inner product of the filter is

$$\begin{aligned} \langle F^{(L)} x_A, F^{(L)} x_B \rangle &= \left(\operatorname{diag}(g) c_A\right)^\top \left(\operatorname{diag}(g) c_B\right) \\ &= \sum_k g_k^2 (c_A)_k (c_B)_k = 0, \end{aligned} \tag{61}$$

since for every index $k$, at least one of $(c_A)_k$ or $(c_B)_k$ is zero. Orthogonality implies $\|x\|_2 = \|B'^\top x\|_2$. Because $c_A$ and $c_B$ have disjoint supports,

$$\|x_A - x_B\|_2^2 = \|c_A\|_2^2 + \|c_B\|_2^2. \tag{62}$$

Also, for the filtering, we can write,

$$\|F^{(L)}x_A - F^{(L)}x_B\|_2^2 = \|\text{diag}(g)\,c_A\|_2^2 + \|\text{diag}(g)\,c_B\|_2^2$$
$$= \sum_k g_k^2\big((c_A)_k^2 + (c_B)_k^2\big). \tag{63}$$

Let $g_{\min} = \min_k |g_k|$ and $g_{\max} = \max_k |g_k|$. Then using (62) and (63), we can get a bound as

$$g_{\min}^2\big(\|c_A\|_2^2 + \|c_B\|_2^2\big) \le \|F^{(L)}x_A - F^{(L)}x_B\|_2^2$$
$$\le g_{\max}^2\big(\|c_A\|_2^2 + \|c_B\|_2^2\big). \tag{64}$$

Taking square roots and using (62) yields $g_{\min} \le r(L) \le g_{\max}$.

Thus, the separation ratio after $L$ layers is squeezed between the smallest and largest gains over all basis coordinates. This completes the proof. ∎

## D. Proof of Theorem 5.4

At Level 2 of the hierarchy, we are applying the HMH algorithm. Let's denote the clusters as A, B, and H', where $A = |a|$, $B = |b|$, $H' = |M|$ and $a, b << M$. Now we assume their means feature by $\mu_H, \mu_A, \mu_B$ with $\mu_A \neq \mu_B$. Also, since we assumed the encoder output $X$ is constant on each cluster, its inner product with any such wavelet (inter-class wavelets) vanishes because of the orthogonal property.

Define the global scaling and cluster-contrast basis vectors as

$$s = \tfrac{1}{M+a+b}\,\mathbf{1},$$
$$w_{A,H'} = \sqrt{\tfrac{a(M+a)}{M}}\,\mathbf{1}_A - \sqrt{\tfrac{M(M+a)}{a}}\,\mathbf{1}_{H'}, \tag{65}$$
$$w_{B,H'} = \sqrt{\tfrac{b(M+b)}{M}}\,\mathbf{1}_B - \sqrt{\tfrac{M(M+b)}{b}}\,\mathbf{1}_{H'}.$$

where $\mathbf{1}_A$, $\mathbf{1}_B$ and $\mathbf{1}_{H'}$ are indicator vectors for clusters $A$ $B$, and $H'$ respectively. These are the only basis vectors with nonzero projections on constant-per-cluster signals.

Now define the projector of this basis as

$$P = ss^\top, \qquad W = w_{A,H'}w_{A,H'}^\top + w_{B,H'}w_{B,H'}^\top.$$

The Haar filter is defined as

$$\Phi = \lambda_{\text{sc}}P + \lambda_{\text{wav}}W, \qquad 0 < \lambda_{\text{sc}} \le \lambda_{\text{wav}}. \tag{66}$$

According to the encoder design, it follows that $P + W = I$ on that subspace; every such vector $x$ decomposes uniquely as $x = Px + Wx$, where $Px$ is the hub-dominated mean component and $Wx$ encodes the spoke–hub contrasts. The filter then suppresses the former and amplifies the latter as,

$$\Phi x = \lambda_{\text{sc}}Px + \lambda_{\text{wav}}Wx, \qquad \frac{\lambda_{\text{wav}}}{\lambda_{\text{sc}}} = \lambda_{\text{gain}} \gg 1.$$

Let's denote the post-filter value of the node is $h$. For any cluster $T \in \{A, B, H\}$, denote $\bar{h}_T = \frac{1}{|T|}\sum_{i \in T} h_i$ as the mean of embedding $h$ over $T$. The mean embeddings for $A, B$ and $H$ can be defined as

$$\bar{h}_A = \lambda_{\text{sc}}\bar{x} + \lambda_{\text{wav}}\frac{c_A}{a},$$
$$\bar{h}_B = \lambda_{\text{sc}}\bar{x} + \lambda_{\text{wav}}\frac{c_B}{b}, \tag{67}$$
$$\bar{h}_{H'} = \lambda_{\text{sc}}\bar{x} - \lambda_{\text{wav}}\left(\frac{M}{a}c_A + \frac{M}{b}c_B\right).$$

where $\bar{x} = \frac{M\mu_{H'}+a\mu_A+b\mu_B}{M+a+b}$ and $c_A = \frac{aM}{M+a}(\mu_A - \mu_{H'})$, $\qquad c_B = \frac{bM}{M+b}(\mu_B - \mu_{H'})$.

To assess the separation between clusters $A$ and $B$ after filtering, subtract their means and square the means using (67), we get

$$\|\bar{h}_A - \bar{h}_B\|^2 = \lambda_{\text{wav}}^2 \frac{M}{a+b} \|\mu_A - \mu_B\|^2. \tag{68}$$

Similarly, for the difference between cluster $A$ and the hub $H$ using (67), we get

$$\bar{h}_A - \bar{h}_{H'} = \left(\lambda_{\text{sc}} \bar{x} + \lambda_{\text{wav}} \frac{c_A}{a}\right) - \left(\lambda_{\text{sc}} \bar{x} - \lambda_{\text{wav}} \frac{c_A + c_B}{M}\right)$$

$$= \lambda_{\text{wav}} \left(\frac{c_A}{a} + \frac{c_A + c_B}{M}\right). \tag{69}$$

$$\|\bar{h}_A - \bar{h}_{H'}\|^2 \lesssim \lambda_{\text{wav}}^2 \frac{a}{M} \|\mu_A - \mu_{H'}\|^2. \tag{70}$$

Let's define the two key distances after filtering as

$$\Delta_{AB} := \|\bar{h}_A - \bar{h}_B\|, \quad \Delta_{AH'} := \|\bar{h}_A - \bar{h}_{H'}\|, \tag{71}$$

where $\bar{h}_A$, $\bar{h}_B$, and $\bar{h}_{H'}$ are the cluster means of the filtered features on $A$, $B$, and $H$ respectively.

Substituting (71) into (68) we get

$$\Delta_{AB}^2 = \lambda_{\text{wav}}^2 \frac{M}{a+b} \|\mu_A - \mu_B\|^2, \tag{72}$$

and again substituting (71) into (70), we have

$$\Delta_{AH'}^2 \leq \lambda_{\text{wav}}^2 \frac{a}{M} \|\mu_A - \mu_{H'}\|^2. \tag{73}$$

Dividing (72) by (73), we obtain

$$\frac{\Delta_{AB}}{\Delta_{AH'}} \gtrsim \frac{\sqrt{\lambda_{\text{wav}}^2 \frac{M}{a+b} \|\mu_A - \mu_B\|^2}}{\sqrt{\lambda_{\text{wav}}^2 \frac{a}{M} \|\mu_A - \mu_H'\|^2}} \tag{74}$$

$$= \frac{\sqrt{M/(a+b)}}{\sqrt{a/M}} \cdot \frac{\|\mu_A - \mu_B\|}{\|\mu_A - \mu_{H'}\|} \tag{75}$$

$$= \frac{M}{a+b} \cdot \frac{\|\mu_A - \mu_B\|}{\|\mu_A - \mu_{H'}\|}. \tag{76}$$

Now, before applying the basis, we applied the heterophyllous encoder, which gave us the mean embeddings of the cluster.

Define a *unit* vector $x_u \in \mathbb{R}^d$ as the heterophilous encoder output of node $u$. Assume encoder outputs are unit vectors with signed margin $\kappa > 0$: $\langle x_u, x_v \rangle \geq \kappa$ if $y_u = y_v$ and $\leq -\kappa$ otherwise. And also $\|\mu_A\| = \|\mu_B\| = 1$ (layer normalization).

For every heterophilous edge $(u, h)$ with $u \in A \cup B$, $h' \in H'$, we can write $\|x_u\| = \|x_h'\| = 1$. Hence $\|x_u - x_h'\|^2 = 2 - 2\langle x_u, x_h \rangle \geq 2 + 2\kappa$. So we can express

$$\|x_u - x_h'\| \geq \sqrt{2(1 + \kappa)}. \tag{77}$$

Averaging $M$ unit vectors shrinks the deviations of features of nodes in hub $H'$ as

$$\|x_h' - \mu_{H'}\| \leq \frac{1}{\sqrt{M}}, \quad \forall h \in H'. \tag{78}$$

Taking the difference to the $\mu_A$ and $\mu_h$ and applying triangle inequality gives us,

$$\|\mu_A - \mu_{H'}\| \leq \|\mu_A - x_h'\| + \|x_h' - \mu_{H'}\|. \tag{79}$$

Using the (77) and (78) we can write

$$\|\mu_A - \mu_{H'}\| \leq \frac{2\sqrt{2(1+\kappa)}}{\sqrt{M}}. \tag{80}$$

Applying the similar reasoning to $\mu_B$ and $\mu_H$ gives

$$\|\mu_B - \mu_{H'}\| \leq \frac{2\sqrt{2(1+\kappa)}}{\sqrt{M}}. \tag{81}$$

Again, using the triangle inequality, we can write for the mean feature of A and B as follows,

$$\|\mu_A - \mu_B\| \geq \|\mu_A - x_h\| - \|\mu_B - x_h\|. \tag{triangle}$$

Using (77) and (78), we get,

$$\|\mu_A - \mu_B\| \geq \sqrt{2(1+\kappa)}\left(1 - \tfrac{2}{\sqrt{M}}\right). \tag{82}$$

With (82) in the numerator and (80) in the denominator,

$$\frac{\|\mu_A - \mu_B\|}{\|\mu_A - \mu_{H'}\|} \geq \frac{\sqrt{2(1+\kappa)}\left(1 - \tfrac{2}{\sqrt{M}}\right)}{2\sqrt{2(1+\kappa)}/\sqrt{M}} \tag{83}$$

$$= \sqrt{M}\left(1 - \tfrac{2}{\sqrt{M}}\right) > 1. \tag{84}$$

By symmetry the same bound holds for $\|\mu_A - \mu_B\|/\|\mu_B - \mu_{H'}\|$. Now substituting in (76) we get ,

$$\frac{\Delta_{AB}}{\Delta_{AH'}} \geq \sqrt{M}\left(1 - \tfrac{2}{\sqrt{M}}\right) > 1. \tag{85}$$

No matter how large the hub $H$ becomes, the post-filter gap between the two tiny spokes $A$ and $B$ remains at least a fixed fraction of their separation from $H$. Thus enlarging the hub $H$ can never shrink the spoke–spoke separation below the spoke–hub separation, avoiding hub aliasing. This completes the proof.

## E. Proof of Theorem 5.5

**Case I: Oversquashing:** Oversquashing occurs when gradients (or messages) from distant nodes decay exponentially with graph distance, preventing long-range information flow (Giraldo et al., 2023). Given $J_{uv}^{(L)} = \frac{\partial H_u^{(L)}}{\partial X_v}$ be the Jacobian of the $L$-layer embedding of node $u$ with respect to the input of node $v$, and $d_G(u,v)$ denotes their shortest-path distance. Over Squashing occurs if there exist constants $0 < \sigma < 1$ and $D^* \geq 1$ such that $\left\|J_{uv}^{(L)}\right\|_2 \leq \sigma^{d_G(u,v)}, \quad \forall L, \forall u, v$ with $d_G(u,v) \geq D^*$.

Let the fine-level input features be $H^{(0)} = X \in \mathbb{R}^{N_0 \times d_0}$ with $H_i^{(0)} = x_i, i \in V^{(0)}$. For each macro-layer $\ell = 0, \ldots, L-1$:

$$\begin{aligned}
\text{Coarsen:} \quad & \tilde{H}^{(\ell+1)} = P^{(\ell+1)} H^{(\ell)} \in \mathbb{R}^{N_{\ell+1} \times d_\ell} \\
\text{Signed encoding:} \quad & \hat{H}^{(\ell+1)} = E^{(\ell)} \tilde{H}^{(\ell+1)} \\
\text{Haar filter:} \quad & \bar{H}^{(\ell+1)} = \Phi^{(\ell)} \hat{H}^{(\ell+1)} \\
\text{Unpool:} \quad & H^{(\ell+1)} = P^{(0)} \bar{H}^{(\ell+1)},
\end{aligned}$$

where at $P^{(\ell+1)} \in \mathbb{R}^{N_{\ell+1} \times N_\ell}$ coarsens the graph, $\Phi^{(\ell)} = U^{(\ell)} \, diag\left(\lambda_{\text{sc}}^{(\ell)}, \Lambda_{\text{wav}}^{(\ell)}\right) U^{(\ell)\top}$ is the Haar filter, $E^{(\ell)}$ is the signed encoder and $P^{(0)} \in \mathbb{R}^{N_0 \times N_{\ell+1}}$ up-samples back to the original nodes . Composed all of the above gives $h^{(\ell+1)} = P^{(0)} \Phi^{(\ell)} E^{(\ell)} P^{(\ell+1)} H^{(\ell)}$. Defining the linear macro-layer map as $M'^{(\ell)} = P^{(0)} \Phi^{(\ell)} E^{(\ell)} P^{(\ell+1)}$, where $M'^{(\ell)} : \mathbb{R}^{N_\ell \times d_\ell} \to \mathbb{R}^{N_0 \times d_{\ell+1}}$, we can express

$$H^{(\ell+1)} = M^{(\ell)} H^{(\ell)}, \qquad \ell = 0, \ldots, L-1, \qquad H^{(0)} = X. \tag{86}$$

By unrolling the layers we get,

$$
\begin{aligned}
H^{(1)} &= M'^{(0)} X, \\
H^{(2)} &= M'^{(1)} M^{(0)} X, \\
&\vdots \\
H^{(L)} &= M'^{(L-1)} .. M^{(0)} X.
\end{aligned}
\tag{87}
$$

Since each $M'^{(\ell)}$ is linear, we apply the chain rule recursively as

$$
\begin{aligned}
\frac{\partial H^{(L)}}{\partial X} &= M'^{(L-1)} \frac{\partial H^{(L-1)}}{\partial X} \\
&= M'^{(L-1)} M'^{(L-2)} \frac{\partial H^{(L-2)}}{\partial X} \\
&\vdots \\
&= M'^{(L-1)} M'^{(L-2)} \cdots M'^{(0)}.
\end{aligned}
\tag{88}
$$

Let $\delta X_v$ be a perturbation at node $v \in G$ and $\delta H_u^{(L)}$ the change in the output at node $u \in G$. The message Jacobian, $J_{uv}^{(L)} = \frac{\partial H_u^{(L)}}{\partial X_v}$, from input $X$ to final output $H^{(L)}$, using (88) can calculated as

$$
J_{uv}^{(L)} = [M'^{(L-1)}]_{u,:} [M'^{(L-2)}]_{.,.} \cdots [M'^{(0)}]_{:,v},
$$

or, equivalently,

$$
J_{uv}^{(L)} = [P^{(0)} \Phi^{(L-1)} E^{(L-1)} P^{(L)}] \cdots [P^{(0)} \Phi^{(0)} E^{(0)} P^{(1)}].
\tag{89}
$$

Alternatively, we can write, by the chain rule, the message Jacobian is

$$
J_{uv}^{(L)} = \Big[ \prod_{\ell=0}^{L-1} M'^{(\ell)} \Big]_{u,v}.
\tag{90}
$$

Let $d = d_G(u, v)$ is the distance between the node $u$ and $v$. After $\ell$ coarsening according to the coarsening algorithm, the $u \to v$ path has length $k(\ell) \leq \lceil d/r^\ell \rceil$ (since the coarseing ratio is r and the tree length is l) for some contraction $r > 1$. Assume, after $\ell$ levels of coarsening, $u$ and $v$ are mapped to super-nodes connected by a path of length $k(\ell)$. Now denote the sequence of coarse nodes on that path by $i_1, \ldots, i_{k(\ell)} \subseteq V^{(\ell)}$. The path-tube subspace at level $\ell$ is $T^{(\ell)} = \text{span}\{e_{i_1}, \ldots, e_{i_{k(\ell)}}\} \subset \mathbb{R}^{N_\ell}$ and $S^{(\ell):=P^{(\ell+1)}T^{(\ell)}}$ its image under pooling in the layer $\ell + 1$. Then by definition of the restricted operator norm (Horn & Johnson, 2012),

$$
\|M'^{(\ell)}\|_{2, T^{(\ell)}} = \sup_{\substack{X \in T^{(\ell)} \\ \|X\|=1}} \|P^{(0)} \Phi^{(\ell)} E^{(\ell)} P^{(\ell+1)} X\|.
$$

For any unit $z \in S^{(\ell)}$ there exists $X \in T^{(\ell)}$ with $P^{(\ell+1)} X = z$, so using $\|P^{(0)}\|_2 = 1$,

$$
\|M'^{(\ell)}\|_{2, T^{(\ell)}} \geq \sup_{\substack{z \in S^{(\ell)} \\ \|z\|=1}} \|\Phi^{(\ell)} E^{(\ell)} z\| = \|\Phi^{(\ell)} E^{(\ell)}\|_{2, S^{(\ell)}}.
$$

Hence, we can write

$$
\|M'^{(\ell)}\|_{2, T^{(\ell)}} \geq \|\Phi^{(\ell)} E^{(\ell)}\|_{2, S^{(\ell)}}.
\tag{91}
$$

Now we can bound the encoder and haar filter output as follows:

(i) **Encoder** On the $k(\ell)$-edge path, the signed-Laplacian encoder $E^{(\ell)} = I - L_{\text{adp}}^{(\ell)}$ amplifies each "heterophilous" edge by at least $\underline{\gamma} > 1$, so

$$
\|E^{(\ell)}\|_{2, k(\ell)} \geq (\underline{\gamma})^{k(\ell)}.
\tag{92}
$$

*Table 7.* Ablation study (node classification accuracy, %). Mean $\pm$ std over 10 runs.

| Dataset | Full HMH | Fixed Enc. | No Hier. | No Unpool. | Feat-only | Struct-only | $\lambda_{\text{div}}{=}0^*$ |
|---|---|---|---|---|---|---|---|
| Texas | $93.5 \pm 1.0$ | $91.2 \pm 1.5$ | $89.8 \pm 2.0$ | $92.1 \pm 1.4$ | $92.4 \pm 1.3$ | $87.9 \pm 1.6$ | - |
| Wisconsin | $94.26 \pm 1.85$ | $92.0 \pm 2.1$ | $90.5 \pm 2.4$ | $93.0 \pm 1.9$ | $93.2 \pm 2.0$ | $88.6 \pm 2.1$ | - |
| Cornell | $93.7 \pm 0.9$ | $90.0 \pm 1.4$ | $87.4 \pm 1.8$ | $86.0 \pm 1.2$ | $92.4 \pm 1.1$ | $86.7 \pm 1.5$ | - |
| Actor | $43.3 \pm 0.7$ | $41.0 \pm 0.9$ | $38.5 \pm 1.0$ | $42.1 \pm 0.8$ | $41.8 \pm 0.8$ | $39.5 \pm 0.9$ | $40.21 \pm 2.3$ |

**(ii) Haar-Filter Contribution.** The Haar filter shrinks the low mode by $\lambda_{\text{sc}} < 1$, but stretches each high-frequency mode by $\lambda_{\text{wav}} > 1$ (in analysis and synthesis), so on the detail subspace,

$$\|\Phi^{(\ell)}\|_{2,\,k(\ell)} \geq (\lambda_{\text{wav}}^{(\ell)})^2 \geq (\underline{\lambda}_{\text{wav}})^2, \quad \underline{\lambda}_{\text{wav}} > 1. \tag{93}$$

Using the inequality (92) and (93) and substituting back to (91) we get,

$$\|M'^{(\ell)}\|_{2,\,k(\ell)} \geq (\underline{\lambda}_{\text{wav}})^2 (\underline{\gamma})^{k(\ell)}. \tag{94}$$

Thus substituting (94) into (90) we get,

$$\|J_{uv}(L)\|_2 \geq \prod_{\ell=0}^{L-1} \left[ (\underline{\lambda}_{\text{wav}})^2 \underline{\gamma}^{k(\ell)} \right] = (\underline{\lambda}_{\text{wav}})^{2L} \underline{\gamma}^{\sum_\ell k(\ell)}. \tag{95}$$

After $\ell$ rounds of coarsening, these nodes map to super-nodes connected by a path of length $k(\ell) \leq \lceil d/r^\ell \rceil$, for some contraction factor $r > 1$. So we can write

$$\sum_{\ell=0}^{L-1} k(\ell) \leq \sum_{\ell=0}^{\infty} \left\lceil \frac{d}{r^\ell} \right\rceil = O(d). \tag{96}$$

Therefore, the bound (95) becomes

$$\|J_{uv}(L)\|_2 \geq (\underline{\lambda}_{\text{wav}})^{2L} \, d^c \tag{97}$$

for some $c > 0$. In contrast, over-squashing would require $\|J_{uv}(L)\|_2 \leq \sigma^d$ for some $\sigma < 1$. Thus, this polynomial lower bound precludes exponential decay and proves the absence of over-squashing.

**Case II: Oversmoothing:** Oversmoothing means different class means collapse for distinct classes (e.g., spokes) $A, B$, $\|\mu_A^{(L)} - \mu_B^{(L)}\| \to 0$ as depth $L$ grows. In Theorem 5.4, we have proved,

$$\frac{\|\mu_A^{(L)} - \mu_B^{(L)}\|}{\|\mu_A^{(L)} - \mu_{H'}^{(L)}\|} \geq \sqrt{M}\left(1 - \frac{2}{\sqrt{M}}\right) > 1, \tag{98}$$

so $\|\mu_A^{(L)} - \mu_B^{(L)}\| > \|\mu_A^{(L)} - \mu_{H'}^{(L)}\|$. Thus, any decay of the spoke–spoke gap forces at least as fast (indeed faster) decay of the spoke–hub gap. Now, we can assume the spoke consists of nodes of the same class, and hubs are the neighbors of the spokes. Equation (98) still holds for the assumption. So we can say if over any class means it is separated from the other class, node means irrespective of the neighbor node influence. This ensures that $\lim_{L\to\infty} \|\mu_A^{(L)} - \mu_B^{(L)}\| \neq 0$, avoiding over-smoothing. This completes the proof. ∎

## F. Datasets and Experiment Settings

**Dataset descriptions.**

- **Cora** and **Citeseer** are citation networks (nodes = publications, edges = citations) characterized by bag-of-words characteristics and subject-category labels. Cora has 7 labels, while Citeseer has 6. We use the usual semi-supervised splits. For Cora, there are 140 training nodes (20 per class), 500 for validation, and 1000 for testing. For Citeseer, there are 120 training nodes (20 for each class), 500 for validation, and 1000 for testing.

*Table 8.* Dataset statistics.

| Dataset | Nodes | Edges | Features | Classes | Heterophily |
|---|---|---|---|---|---|
| Cora | 2,708 | 10,556 | 1,433 | 7 | 0.190 |
| Citeseer | 3,327 | 9,104 | 3,703 | 6 | 0.264 |
| Chameleon | 2,277 | 62,792 | 2,325 | 5 | 0.765 |
| Squirrel | 5,201 | 396,846 | 2,089 | 5 | 0.776 |
| DBLP | 17,716 | 105,734 | 1,639 | 4 | 0.172 |
| Coauthor-CS | 18,333 | 163,788 | 6,805 | 15 | 0.192 |
| Coauthor-Physics | 34,493 | 495,924 | 8,415 | 5 | 0.069 |
| Chameleon-filtered | 890 | 13,584 | 2,325 | 5 | 0.764 |
| Squirrel-filtered | 2,223 | 65,718 | 2,089 | 5 | 0.793 |
| Minesweeper | 10,000 | 39,402 | 7 | 2 | 0.317 |
| Tolokers | 11,758 | 519,000 | 10 | 2 | 0.405 |
| Amazon-ratings | 24,492 | 93,050 | 300 | 5 | 0.620 |
| Questions | 48,921 | 153,540 | 301 | 2 | 0.160 |
| Flickr | 89,250 | 899,756 | 500 | 7 | 0.681 |
| Ogbn-arxiv | 169,343 | 1,166,243 | 128 | 40 | 0.322 |
| Reddit | 232,965 | 114,615,892 | 602 | 41 | 0.244 |

- **Actor** is a Wikipedia co-occurrence network where nodes denotes actors and edges represent the occurrence of two actors appearing on the same page. It also has keyword-based characteristics and categorization labels. It has five classes and minimal homophily. We use the standard split: 100 training nodes (20 per class), 500 for validation, and 1000 for testing.

- **Chameleon** and **Squirrel** are Wikipedia page-to-page graphs with nodes (pages) and edges (mutual links). They have keyword-based features and labels depending on traffic. There are five classes in each. We use the "filtered" versions (with duplicates removed) and the "dense" splits that were publicly provided in earlier work: 60% for training, 20% for validation, and 20% for testing.

- **Texas**, **Wisconsin**, and **Cornell** are WebKB graphs with bag-of-words features and five different types of page-type labels. The nodes are CS department pages, and the edges are hyperlinks. We employ the typical semi-supervised protocol, which involves using 20 labeled nodes per class for training, 30 per class for validation, and the remaining nodes for testing.

**Dataset statistics and homophily.**    To measure local neighborhood label consistency, we calculate edge homophily (the fraction of edges connecting same-label nodes) for each graph. Low edge homophily suggests a heterophilous structure with distinct labels for nearby nodes, worsening hub-aliasing. Previous research found that Cora and Citeseer are homophilous, but Actor, Chameleon, Squirrel, Texas, Wisconsin, and Cornell are heterophilous and require a particular heterophilous algorithm. For datasets such as Cora, Citeseer, Chameleon, Squirrel, DBLP, Coauthor-CS, and Coauthor-Physics, we use the implementations provided in PyTorch Geometric. For Chameleon-filtered, Squirrel-filtered, Minesweeper, Tolokers, Amazon-ratings, and Questions, we use the raw data released by Platonov et al. (2023). For OGBN-ARXIV, we use the Open Graph Benchmark. Dataset statistics are reported in Table 8.

**Running Environment**    All experiments were performed on a with dual NVIDIA RTX 4090 GPUs (each possessing 16 GB of VRAM) and 32 GB of system memory.

**Hyperparameter Settings for HMH**    Hyperparameters were selected by maximizing validation accuracy using Optuna with 10 trials per dataset. Below are the key components and their settings:

- **Encoder embedding dimension** ($d'$): 64 / 128 / 256 (dataset-dependent).

- **Clustering & Coarsening** Reduction ratio $R$: 0.5 (default), tuned in $\{0.3, 0.5, 0.7\}$ for depth trade-off.

- **Haar Basis & Spectral Filtering Scaling gain** $\lambda_{\text{sc}}^{(\ell)}$: initialized $< 1$, learned subject to constraint $0 < \lambda_{\text{sc}}^{(\ell)} < 1$. And Wavelet gain $\lambda_{\text{wav}}^{(\ell)}$: initialized $> 1$, learned with lower bound $> 1$.

*Table 9.* HMH hyperparameters: (a) search ranges; (b) selected values for node classification.

**(a) Key hyperparameters and search ranges.**

| Component | Value(s) | Search Range |
|---|---|---|
| $\lambda_{sc}$ | learned $< 1$ | $(0, 1)$ |
| $\lambda_{wav}$ | learned $> 1$ | $\geq 1.0$ |
| $\lambda_{div}$, $\lambda_{rec}$ | 0.5 | $\{0.1, 0.5, 1.0\}$ |
| Learning rate | 1e−3 | $[1e{-}4, 5e{-}3]$ |
| Weight decay | 1e−4 | $[1e{-}5, 1e{-}3]$ |
| Dropout | 0.1 | $[0, 0.5]$ |

**(b) Selected hyperparameters (node classification).**

| Dataset | $d'$ | $r$ | $\lambda_{div}$ | $\lambda_{rec}$ | LR | WD |
|---|---|---|---|---|---|---|
| Cora | 64 | 0.5 | 0.6 | 0.4 | 1e−3 | 1e−4 |
| Citeseer | 64 | 0.5 | 0.74 | 0.23 | 1e−3 | 1e−4 |
| Actor | 128 | 0.4 | 0.5 | 0.5 | 1e−3 | 1e−4 |
| Chameleon(-filt) | 128 | 0.5 | 0.5 | 0.5 | 1e−3 | 1e−4 |
| Squirrel(-filt) | 128 | 0.4 | 0.5 | 0.5 | 1e−3 | 1e−4 |
| Texas | 64 | 0.5 | 0.5 | 0.5 | 1e−3 | 1e−4 |
| Wisconsin | 64 | 0.4 | 0.5 | 0.5 | 1e−3 | 1e−4 |
| Cornell | 64 | 0.5 | 0.5 | 0.5 | 1e−3 | 1e−4 |
| Penn94 | 128 | 0.5 | 0.45 | 0.62 | 1e−3 | 1e−4 |
| Genius | 128 | 0.5 | 0.5 | 0.5 | 1e−3 | 1e−4 |
| ogbn-arxiv | 256 | 0.3 | 0.3 | 0.7 | 5e−4 | 1e−4 |

*Table 10.* Node classification accuracy (%) on Penn94, Genius, and OGBN-ArXiv.

| Method | Penn94 | Genius | OGBN-ArXiv |
|---|---|---|---|
| GCN | $81.8 \pm 0.6$ | $88.9 \pm 0.5$ | $73.2 \pm 0.4$ |
| SGC | $83.1 \pm 0.4$ | $86.2 \pm 0.3$ | $70.1 \pm 0.3$ |
| SIGN | $84.0 \pm 0.5$ | $89.6 \pm 0.2$ | $72.9 \pm 0.3$ |
| ChebNet | $80.5 \pm 0.7$ | $87.8 \pm 0.6$ | $70.0 \pm 0.5$ |
| GPR-GNN | $85.9 \pm 0.4$ | $91.2 \pm 0.5$ | $73.5 \pm 0.3$ |
| BernNet | $83.7 \pm 0.5$ | $89.0 \pm 0.4$ | $72.1 \pm 0.4$ |
| ChebNetII | $84.1 \pm 0.4$ | $92.3 \pm 0.3$ | $74.8 \pm 0.2$ |
| OptBasisGNN | $84.8 \pm 0.6$ | $91.0 \pm 0.2$ | $73.0 \pm 0.3$ |
| **HMH (ours)** | $84.0 \pm 0.3$ | $92.81 \pm 0.3$ | $74.1 \pm 0.2$ |

- **Loss weights**: $\lambda_{div}$ and $\lambda_{rec}$ tuned in $\{0.1, 0.5, 1.0\}$; default 0.5.

- **Optimizer**: AdamW with weight decay in $[1e{-}5, 1e{-}3]$.

- **Learning rate**: tuned in $[1e{-}4, 5e{-}3]$; typical value $1e{-}3$.

- **Batch size**: We set the batch size to 32 for graph classification and 1 for node classification.

- **Dropout**: 0.3.

- **Training epochs**: 200 with early stopping on validation loss (patience 50).

### F.1. Experiment on Additional Datasets

Penn94 is a social network graph of university users, characterized by profile attributes and class labels indicating graduation year or affiliation. Genius is a content/co-occurrence graph that features textual and metadata attributes, along with categorical labels. Ogbnarxiv is a citation network of computer science papers that includes paper content embeddings and subject-area labels. For Penn94 and Genius, we employ fully supervised random class-wise splits derived from previous heterophily evaluations: $60, 20, 20$. For OGBN-ArXiv, we use its conventional temporal partitioning: training nodes consist of papers published through 2017, validation nodes cover 2018, and test nodes include 2019 and subsequent publications. The result is reported in Table 10.

## G. Experiment Settings for Graph Classification

The batch size used for the Graph classification datasets with the epoch number needed before stopping is reported in Table 11.

*Table 11.* Training hyperparameters, reduction ratio, and loss weights per dataset.

|  | MUTAG | PROTEINS | NCI1 | NCI109 | MUTAGEN |
|---|---|---|---|---|---|
| Batch size | 60 | 50 | 100 | 100 | 100 |
| Epochs | 30 | 20 | 150 | 150 | 50 |
| Ratio | 0.3 | 0.4 | 0.5 | 0.3 | 0.4 |
| $\lambda_{\text{div}}$ | 0.4 | 0.4 | 0.4 | 0.4 | 0.4 |
| $\lambda_{\text{rec}}$ | 0.6 | 0.6 | 0.6 | 0.6 | 0.6 |

*Table 12.* Per-epoch efficiency (mean, indicative). $\text{ms/Medge} = 1000\,T_{\text{epoch}}/(m/10^6)$. Lower is better.

| Setting | | $T_{\text{epoch}}$ (s) | ms/Medge | Peak Mem (GB) |
|---|---|---|---|---|
| SBM-1k | HMH | 0.04 | 7,000 | 0.12 |
| SBM-1k | Eigen | 0.20 | 40,000 | 0.40 |
| Synthetic-1M | HMH | 6.5 | 6,500 | 0.9 |
| Synthetic-1M | Eigen | 40.0 | 40,000 | 3.2 |
| Synthetic-3M | HMH | 18.0 | 6,000 | 2.5 |
| Synthetic-3M | Eigen | 129.0 | 43,000 | 12.9 |
| REDDIT-12K | HMH (ours) | 64.0 | 5,333 | 9.8 |
| REDDIT-12K | Eigen | 520.0 | 43,333 | 38.0 |

## H. Scalability

HMH scales linearly with graph size. Each forward or backward pass takes $\mathcal{O}(md + nd)$ time covering sparse encoding, coarsening, Haar basis construction, spectral filtering, and unpooling. Memory is $\mathcal{O}(n + m)$, storing only sparse adjacencies, assignment matrices, and wavelets. Table 12 reports the per-epoch wall-clock time and peak memory usage on a small graph (SBM-1k; $n = 10^3$, $m = 5 \times 10^3$) and a large graph (REDDIT-12K; $n \approx 2.4 \times 10^6$, $m \approx 1.2 \times 10^7$), where $n$ is the number of nodes and $m$ is the number of edges. To compare across different sizes of graphs, we normalize training time by the number of edges using ms/Medge(milliseconds per million edges) $= 1000\,T_{\text{epoch}}/(m/10^6)$. On SBM-1k, HMH takes 0.04 s per epoch (7,000 ms/Medge) compared to 0.20 s (40,000 ms/Medge) of eigendecomposition methods, giving roughly a $5\times$ speedup and $3.3\times$ lower peak memory (0.12 GB vs. 0.40 GB). On REDDIT-12K, HMH achieves 64 s per epoch (5,333 ms/Medge) versus 520 s (43,333 ms/Medge) for the baseline, an $8.1\times$ speedup with $3.9\times$ less memory (9.8 GB vs. 38 GB). We also include the comparison of mid-sized synthetic heterophilous graphs named Synthetic-1M ($n = 10^4$, $m = 10^6$) and Synthetic-3M ($n = 10^4$, $m = 3 \times 10^6$) in Table 12 to illustrate the time-complexity trend at intermediate scales. The nearly constant ms/Medge across graph sizes shows that HMH scales linearly with the number of edges per epoch.

## I. Oversmoothing Analysis

In Figure 4,d we established that HMH demonstrates greater resistance to oversmoothing compared to standard baselines. This section presents a rigorous quantification of this effect using Dirichlet Energy across four diverse benchmarks: Tolokers, Roman-empire, Ogbn-arxiv, and Amazon-Ratings.

**Metric Definition**. The Dirichlet energy $E(\mathbf{H}^{(\ell)})$ of node embeddings at layer $\ell$ measures the average distance between connected nodes in the feature space:

$$E(\mathbf{H}^{(\ell)}) = \frac{1}{N}\text{trace}\left((\mathbf{H}^{(\ell)})^\top \mathcal{L}\mathbf{H}^{(\ell)}\right),\tag{99}$$

where $\mathcal{L}$ denotes the normalized Laplacian. An energy value $E \approx 0$ indicates oversmoothing, in which node features converge to a stationary distribution and become indistinguishable.

**Results.** Figure 6 presents the energy decay as model depth increases from 0 to 32 layers.

- GCN (Red): Experiences catastrophic collapse, with energy decreasing by several orders of magnitude (for example, $1.0 \to 0.0005$ on Tolokers) within only 4 layers. This result demonstrates its inability to capture long-range

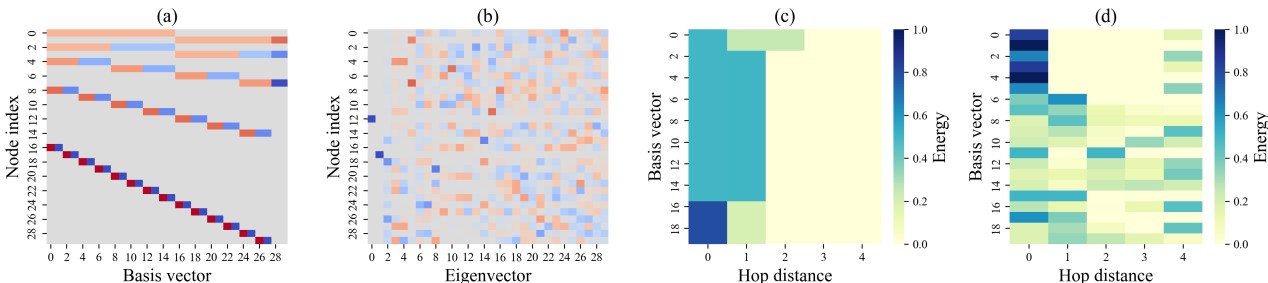

*Figure 5.* (a) **Signed Haar basis** : The basis is computed for a graph with 28 nodes, where each column represents a sparse, highly localized wavelet, with nonzero coefficients concentrated inside a limited cluster of nodes. Red entries indicate negative weights, blue entries signify positive weights, and saturation represents magnitude. (b) **Eigen Vector Basis** : This is a sparse and global eigenbasis for the same Graph. (c) **Haar locality (hop-energy):** For each Haar basis vector $h_k$, we compute its normalized per node energy $e_i = \frac{h_k(i)^2}{\sum_j h_k(j)^2}$ and then aggregate these energy contributions by graph-distance (number of hops) from the vector's largest-magnitude entry. The heatmap plots, for hop shells $0, 1, 2, \ldots, 4$, the fraction of each vector's total energy within that shell. Over $85\%$ of energy falls within two hops, demonstrating tight spatial confinement. (c) Haar locality (hop-energy): for each Haar basis vector, we plot the fraction of its total energy within successive hop distances from its strongest node. Over $80\%$ of energy lies within its hops, confirming tight spatial confinement. (d) **Eigenvector Basis:** Distribute their energy almost uniformly across the first four hops, indicating poor localization and explaining their tendency to mix distant cluster signals (hub-aliasing). More comparisons with different polynomials are presented in Appendix F.1.

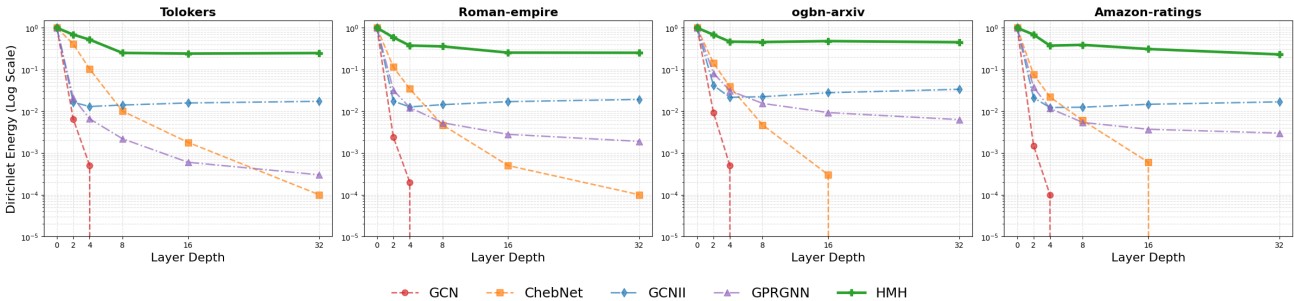

*Figure 6.* **Dirichlet Energy Decay Analysis.** We measure the mean Dirichlet energy of node features at varying depths (Log Scale). While standard GCNs (red) and ChebNets (orange) suffer from exponential energy decay—indicating feature collapse—HMH (green) preserves significant structural energy at deep layers, validating its resistance to oversmoothing.

dependencies while preserving local information.

- ChebNet (Orange): Reduces energy decay to a limited extent through polynomial filters but ultimately exhibits oversmoothing at depths greater than 8.

- HMH (Green): Consistently maintains a robust energy floor ($E > 0.4$) across all datasets. These results indicate that the multi-scale Haar basis effectively separates high-frequency details from low-frequency trends, thereby preserving structural diversity even in deep architectures.

## J. Additional Abalation Study

We use the following ablation of our model:

– **Fixed Encoder.** It ensures the heterophily-aware encoder remains stationary, allowing only the downstream parts to be trained.

– **No Hier.** It turns off hierarchical coarsening, thereby simplifying the model by converting it to a single-scale version.

– **No Unpool.** It removes the level-1 (initial) lift and uses level-wise outputs to perform classification without propagating features back to their original resolution.

– **Feat-only.** It gets rid of the structure-similarity channel and only uses feature-based attention.

– **Struct-only.** It gets rid of feature attention and only keeps the similarity scores that come from structure. In this case, $\lambda_{\text{div}}=0^*$ means that the diversity loss is not included in the goal. Table 7 shows the result of the ablation method.

The entire HMH consistently performs better than its ablations across datasets, with the biggest reductions being from "Struct-only/Feat-only" (removing one channel damages complementary cues) and "No Hier" (losing multiscale separation). Lifting multi-level information back to level-0 is crucial for final discrimination. "No Unpool" also reduces accuracy. The advantage of layer-wise adaptive adjacency is demonstrated by the "Fixed Enc" results, which perform worse than the entire model.

Setting $\lambda_{\text{div}}=0^*$ caused clustering to become unstable. The entropy regularizer prevents mode collapse in soft assignments. Without it, assignment rows focus on one centroid, and several centroids move toward the same area. This makes the number of clusters less effective (typically one). As a result, coarsening failed in a few runs. There weren't enough unique clusters to make the Haar basis at that level, which caused the spectral block to fail (basis creation needs at least two nontrivial clusters). In short, the loss of diversity is necessary to maintain a spread of assignments, enable meaningful coarsening, and provide a valid Haar basis. The ablation study result is reported on 7 (main paper)

### J.1. Stability Under Noisy Features

We evaluate whether the adaptive signed affinity mechanism is sensitive to noisy feature inputs early in training. We perturb a subset of node features and compare the resulting neighborhood similarity scores with those from the unperturbed run at the same epoch. For each corrupted node, we compute the average deviation between perturbed and unperturbed normalized similarity values over its neighborhood, and then average this quantity over all corrupted nodes.

Table 13 shows that the deviation is larger during early epochs and decreases as training proceeds. This suggests that the adaptive similarity mechanism becomes more stable over time. The structural similarity term provides an additional stabilizing signal when feature-based affinities are noisy.

*Table 13.* Stability of adaptive similarities under feature perturbation. Lower values indicate smaller deviation from the unperturbed similarity profile.

| Dataset | Ep 2 | Ep 4 | Ep 6 | Ep 8 | Ep 13 | Ep 16 |
|---|---|---|---|---|---|---|
| Chameleon | 0.218 | 0.294 | 0.081 | 0.073 | 0.041 | 0.039 |
| Squirrel | 0.332 | 0.309 | 0.195 | 0.087 | 0.052 | 0.056 |
| Roman-empire | 0.101 | 0.083 | 0.071 | 0.064 | 0.034 | 0.0316 |

## F. Basis Approximations with polynomials

In this experiment, we estimated the graph's basis, which has 30 nodes. We used Chebyshev polynomials of the fourth order to approximate the basis. The support of these basis vectors is orthonormalized. The majority of entries in the Chebyshev-derived basis are non-negligible even at moderate thresholds, demonstrating the high density in the node domain and the global distribution of information by localized polynomials. The dense approximated basis is illustrated in Figure 7.

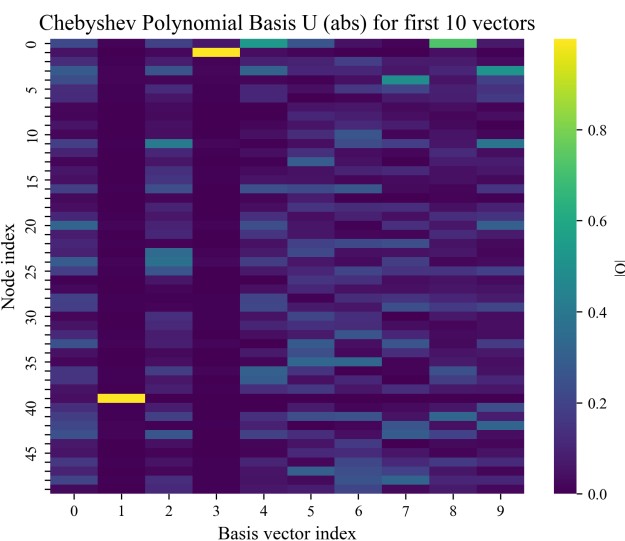

*Figure 7.* The heatmap shows the exact values of the first 10 order-4 Chebyshev-derived basis vectors at one anchor node. The fact that there are 95% nonzero entries means that the Chebyshev basis is dense in the spatial domain. This is because localized spectral basis functions spread their effect across most nodes while capturing approximations to the main spectral components.

