# OpenReview forum: "Hierarchical Multi Scale Graph Neural Networks: Scalable Heterophilous Learning with Oversmoothing and Oversquashing Mitigation"
_ICML.cc/2026/Conference — ICML 2026 regular_

### Official Review · Reviewer_xFoR · 2026-03-05

**Soundness:** 3
**Presentation:** 2
**Significance:** 3
**Originality:** 3
**Overall Recommendation:** 4
**Confidence:** 4

**Summary:**

This paper proposes HMH, a hierarchical multi-scale spectral GNN designed for learning on heterophilous graphs. It first learns adaptive signed affinities from node features and local structure, then builds a soft coarsening hierarchy and constructs sparse, orthonormal Haar wavelet bases at each level to perform diagonal spectral filtering. The multi-scale filtered representations are fused back into the original graph via skip-connected unpooling to mitigate hub domination, oversmoothing, and oversquashing while remaining scalable. Experiments on node- and graph-classification benchmarks show good results.

**Compliance With Llm Reviewing Policy:**

Affirmed.

**Final Justification:**

The authors have basically addressed my concerns, and I am willing to raise my score.

**Key Questions For Authors:**

1. Regarding Theorem 5.1, the proof seems to attribute SMP sign flipping to the strong condition S^2 = I, which generally does not hold for real signed adjacency matrices. This raises concerns that the argument may be overly simplified. Could the authors clarify the exact assumptions and scope under which the claim holds?
2. While hierarchical coarsening can shorten effective path lengths, it may also compress distant information into a small number of supernodes, potentially causing information loss. How does the method perform on more general long-range tasks beyond the provided benchmarks?
3. The proposed $U^{(\ell)}$ is not the eigenbasis of the graph Laplacian (as in classical spectral convolution), but an orthonormal wavelet-like basis constructed from the hierarchy. What is the precise relationship between the learned diagonal gains $\Lambda^{(\ell)}$ (i.e., the “frequency channels” in this basis) and Laplacian-defined frequencies?

**Limitations:**

yes

**Strengths And Weaknesses:**

**Strengths**

1. The paper targets key failure modes in heterophilous learning, including hub domination, oversmoothing, and oversquashing of spectral polynomial bases.
2. It outperforms strong baselines across multiple datasets and emphasizes near-linear implementation with scalable training and inference.

**Weaknesses**

1. The Haar basis construction hardens the soft assignments via an argmax operation, which introduces discontinuities and non-differentiability during training.
2. Negative edges are induced by "low similarity," but low similarity can arise not only from heterophily, but also from noise, missing features, or outliers.
3. The intra-cluster wavelets are defined by splitting each cluster into the "first (r) nodes" and the remaining nodes, but the paper does not specify a canonical ordering rule. If the ordering is arbitrary, the resulting representation may fail to be permutation equivariant.

---

> ### Author Rebuttal · Authors · 2026-03-30
>
> We thank the reviewer for the insightful comments. We address the concerns below.
>
> Weakness 1.
>
> Ans. We agree that the argmax in Eq. 8 introduces a discontinuity. In the revised version, we will replace this step with soft assignment matrices. However, all other steps, including graph clustering, feature coarsening, and cluster basis construction, already use the soft assignment matrix $A_S^{(\ell)}$. To assess the effect of the hardening step, we performed an additional ablation where the hardened (argmax) $\tilde A_S$ in Eq. (8) was replaced directly by soft $A_S^{(\ell)}$. The performance remained essentially unchanged, and on several datasets the soft variant was slightly better (e.g., +0.75\% on Roman Empire, +0.80\% on Chameleon, and +0.54\% on Tolokers). This indicates that the gains of HMH do not depend on argmax hardening. In the revision, we will add the result of this ablation.
>
> Weakness 2
>
> Ans. Our formulation reduces the influence/effects of noisy features by computing the adaptive similarity in Eq. (2) from both feature affinity and structural similarity before applying the neighborhood softmax. So, the model does not rely solely on raw features: the structural term provides an additional signal that helps refine and stabilize similarity estimates when features are noisy, incomplete, or unreliable. In response to Reviewer wAx9's Key Question 1, we also reported the epoch at which the structural term stabilizes the similarity score that arises from the noise. Please refer to the table in that response.
>
> Weakness 3.
>
> Ans. For a cluster $C_k^{(\ell)}$ with $n_k$ nodes, the intra-cluster wavelets are constructed from all split points $r=1,\ldots,n_k-1$ incrementally, so they span a family of local contrast directions rather than a single ordering-dependent direction. Let $P$ be a permutation matrix acting within the cluster. Since each wavelet in Eq.9 is a linear combination of subset indicator vectors, permuting the node order simply permutes these indicators and hence the corresponding wavelets. In particular, for each split $r$, we have $\hat w_{intra,r}=P^T w_{intra,r}$, and thus $\hat W_{intra}=P^T W_{intra}$ (up to a column reordering). Therefore, with $\hat{X}=P^T X$, the filtered output satisfies $\hat W_{intra}\hat W_{intra}^T\hat{X}=P^T W_{intra}W_{intra}^T X$, showing that the operation is permutation equivariant. In the revision, we will include this as a lemma.
>
> Key Question 1.
>
> Ans. We agree that the condition $S^2=I$ is strong for a general signed graph; the statement was an unintended typo. We intended to write that after two-hop propagation, the sign pattern becomes fully positive. The condition is $S^2 = c.1_{n \times n}$, where $c$ is the magnitude of the similarity weights. For example, let $n=2$ and  $S=[-1  -1, -1  -1]$  a $2\times2$ signed matrix; then $S^2=c[1\ 1,1\    1]$, which is a $1_{n \times n}$ matrix and $c=2$.. Thus, the first propagation is fully negative, while the second becomes fully positive. Given the corrected condition, the assumption and scope of Theorem 5.1 remain unchanged, and we will correct this typo in the revision.
>
> Key Question 2
>
> Ans. HMH does not compress the original graph signal. Instead, it provides an additional path for long-range interaction and fuses this information back with the full-resolution features. Filtering is first performed on the original graph, while coarser levels provide additional multiscale context through supernodes. As a result, the model preserves local detail while also enabling long-range interaction through coarse structure. We additionally evaluated HMH on the standard long-range benchmarks Peptides-func and Peptides-struct [1]. The evaluation shows strong evidence of HMH's long-range capabilities. Results are reported both in response to reviewer wAx9 key question 2 and in the link below.
>
> Key Question 3
>
> Ans. The learned basis $U^{(\ell)}$ is a hierarchy-induced orthonormal surrogate basis. In Theorem 5.2, we relate it to initial  $k$ Laplacian frequency ($\lambda_k$) through the adaptive Laplacian $L_{{adp}}^{(\ell)}=L+\Delta^{(\ell)}$. The theorem shows, given the new learned frequency $\mu_k$, the adaptive update approximately preserves the low-frequency band as $\mu_k^{(\ell)} \le \lambda_k + \epsilon_\ell \lambda_\tau$, while it lifts the high-frequency band as $\mu_k^{(\ell)} \ge \lambda_k + \beta_\ell \lambda_{\tau+1}$. Since $U^{(\ell)}$ is constructed from assignments derived from this adaptive structure, the same frequency-separation principle carries over to the hierarchy channels. Therefore, the diagonal gains can be interpreted as learnable coefficients that amplify or suppress the hierarchy-induced low- and high-frequency components of $L_{adap}$. We also provide a plot of the learned gains against the Effective Laplacian frequency in the anonymous link below.
>
> Reference [1]. Vijay Dwivedi et al., Long Range Graph Benchmark
>
> Anonymous Link: https://github.com/graph321/Filter_Plot/blob/main/README.md

---

> > ### Author Rebuttal · Reviewer_xFoR · 2026-04-03
> >
> > The authors have basically addressed my concerns, and I am willing to raise my score.

---

### Official Review · Reviewer_wAx9 · 2026-03-11

**Soundness:** 3
**Presentation:** 3
**Significance:** 3
**Originality:** 3
**Overall Recommendation:** 4
**Confidence:** 3

**Summary:**

The paper introduces Hierarchical Multi-view HAAR (HMH), a novel spectral graph-learning framework designed to tackle the limitations of existing Graph Neural Networks (GNNs) on heterophilous graphs.  To address this, HMH employs an adaptive encoder that learns structure-aware signed affinities—assigning positive weights to similar nodes and negative weights to dissimilar ones—which prevents the progressive erasure of high-frequency contrasts. These embeddings guide the construction of a hierarchical graph structure, where a sparse, orthonormal, and locality-aware Haar basis is built at each level to apply precise spectral filters in near-linear time. Finally, skip-connection unpooling layers reintegrate the multi-scale signals back into the original graph, effectively mitigating hub domination and information bottlenecks.

**Compliance With Llm Reviewing Policy:**

Affirmed.

**Final Justification:**

Concerns are addressed. I will keep the score.

**Key Questions For Authors:**

1. The paper claims the adaptive encoder acts as both a high-pass and low-pass filter simultaneously by dynamically updating edge signs. How sensitive is this mechanism to extremely noisy feature inputs early in the training process before the structural similarity scores stabilize?

2. HMH mitigates oversquashing by logarithmically reducing path lengths through its hierarchical tree. At what depth ($L$) does the information bottleneck begin to re-emerge, if at all, when unpooling signals back to the finest resolution on massive graphs?

**Limitations:**

Limitations are not discussed in the paper.

**Strengths And Weaknesses:**

### Strengths

1. The paper introduces a compelling approach using an adaptive heterophilous encoder that assigns positive weights to homophilous edges and persistent negative weights to heterophilous edges. This effectively resolves the signal-signed flipping and cancellation issues present in existing signed message-passing models.

2. The proposed HMH model demonstrates consistent improvements across benchmarks, achieving up to a 3% performance gain on node classification datasets and up to 7% on graph classification datasets compared to state-of-the-art spectral baselines.

### Weaknesses

1. The current formulation relies on a static graph structure. Adapting this framework to dynamic environments where edges change frequently would require continuously updating the hierarchy over time, which may negate the scalability benefits.

---

> ### Author Rebuttal · Authors · 2026-03-31
>
> We thank the reviewer for the thoughtful comments and suggestions. We address the concerns below.
>
> Weakness 1
>
> Ans. We thank the reviewer for this important observation. The current paper focuses on the static-graph settings only, and we note in the limitations section that adapting directly dynamic graphs are currently a limitation of HMH, and that incorporating dynamic graph learning capabilities into HMH is a future direction.
>
> However, we would like to clarify that extending HMH to dynamic environments would not necessarily negate its scalability benefits. The main reason is that our framework does not rely on recomputing a global Laplacian eigendecomposition. Instead, the hierarchy is built from local and sparse quantities as described in sections 5.1, 5.2, and 5.3. Therefore, when graph changes are local, only the affected neighborhoods, assignment weights, and corresponding coarse edges need to be updated, rather than rebuilding the full hierarchy from scratch. In this sense, the framework is compatible with incremental local updates and scalable to large graphs. In the revised version, we will include the above explanation in the limitations section to clarify that the dynamical graph implementation is out of scope for the current version of HMH.
>
> Key Question 1.
>
> Ans. We thank the reviewer for this important comment. The signed adjacency in Eq. (8) is updated iteratively, so noisy edge signs estimated early in training are not fixed permanently and can be corrected as the node representations become more informative. To quantify this sensitivity, we conducted additional experiments in which we introduced controlled perturbations to a subset of node features and compared their resulting similarity profiles with those of the corresponding unperturbed nodes at the same training epoch. Specifically, for each corrupted node $i$, we measure the average deviation of its neighborhood similarity values as $\Delta S^{(e)}(i)=(1/|N(i)|)\sum_{j\in N(i)}|S_{ij}^{(e,clean)}-S_{ij}^{(e,noisy)}|$, where $S_{ij}^{(e,clean)}$ and $S_{ij}^{(e,noisy)}$ denote the normalized similarity values between nodes $i$ and $j$ at epoch $e$ in the unperturbed and perturbed settings, respectively. We then average this quantity over all perturbed nodes to obtain the dataset-level stability measure: $\Delta_{\text{avg}}^{(e)}=(1/|V_{corr}|)\sum_{i\in V_{corr}}\Delta S^{(e)}(i)$, where $V_{corr}$ is the set of corrupted nodes.
>
> As shown below, on Chameleon, Squirrel, and Roman-empire, this deviation is larger in the early epochs and gradually decreases as training progresses. This indicates that the adaptive similarity is initially more sensitive to noisy features but becomes increasingly stable over time. The observed trend supports our claim that the structural similarity term acts as a stabilizing signal when the feature-based component is noisy in the early stage of training.
>
> Dataset        | Ep 2  | Ep 4  | Ep 6  | Ep 8  | Ep 13 | Ep 16
> ---------------|-------|-------|-------|-------|-------|------
> Chameleon      | 0.218 | 0.294 | 0.081 | 0.073 | 0.041 | 0.039
> Squirrel       | 0.332 | 0.309 | 0.195 | 0.087 | 0.052 | 0.056
> Roman-empire   | 0.101 | 0.083 | 0.071 | 0.064 | 0.034 | 0.0316
>
> Key Question 2
>
> Ans. We thank the reviewer for this important question. In HMH, the information bottleneck does not re-emerge during unpooling in the same way as in standard message passing, because unpooling is not another deep edge-wise propagation process on the original graph. Instead, the final node representation is obtained by directly fusing features from all levels of the hierarchy via skip-connected additive aggregation. Thus, information is not forced to traverse a long sequence of local graph hops when returning to the finest resolution.
>
> Empirically, we did not observe a clear hierarchy depth $L$ at which oversquashing reappears during unpooling. In practice, moderate depths (typically 3 to 4 levels) were sufficient and gave the best performance. In response to key question 2 of reviewer xFoR, we reported the performance of HMH on long- range datasets. We evaluated our method on standard long-range benchmarks, Peptides-func and Peptides-struct (large-scale graphs) [1]. The table is reported below:
> | Model | Peptides-func (Test AP ↑) | Peptides-struct (Test MAE ↓) |
> |---|---:|---:|
> | GCN | 0.5930 ± 0.0023 | 0.3496 ± 0.0013 |
> | GCNII | 0.5543 ± 0.0078 | 0.3471 ± 0.0010 |
> | GINE | 0.5498 ± 0.0079 | 0.3547 ± 0.0045 |
> | GatedGCN | 0.5864 ± 0.0077 | 0.3420 ± 0.0013 |
> | Transformer+LapPE | 0.6326 ± 0.0126 | 0.2529 ± 0.0016 |
> | **HMH (ours)** | **0.622 ± 0.115** | **0.235 ± 0.0023** |
>
> From the above table, we observe HMH performs competitively with the Transformer model and performs better than the baseline Graph Neural network model suggesting that it effectively preserves long-range information rather than losing it during hierarchical coarsening.

---

> > ### Author Rebuttal · Reviewer_wAx9 · 2026-04-01
> >
> > I don't have follow-up questions and will maintain my score.

---

> > > ### Author Response · Authors · 2026-04-01
> > >
> > > Thank you for your comment and suggestion.

---

### Official Review · Reviewer_LyoK · 2026-03-13

**Soundness:** 3
**Presentation:** 3
**Significance:** 4
**Originality:** 4
**Overall Recommendation:** 4
**Confidence:** 3

**Summary:**

This paper proposes HMH (Hierarchical Multi-scale Haar) for node and graph classification, with emphasis on heterophilous graphs and on mitigating hub domination, oversmoothing, and oversquashing. The model first learns an adaptive signed adjacency from feature affinity and structural similarity, then recursively builds a soft graph hierarchy via clustering and coarsening. At each level, it constructs a localized Haar basis and applies diagonal spectral filtering in that basis. The paper provides several theoretical statements intended to justify why the adaptive signed propagation avoids sign flipping and why the hierarchical Haar basis mitigates hub domination.

**Compliance With Llm Reviewing Policy:**

Affirmed.

**Final Justification:**

Taking into account the authors’ rebuttal as well as the feedback from other reviewers, I have chosen to keep my initial rating unchanged at 4.

**Key Questions For Authors:**

1. In "Table 3", the degree thresholds used to define spokes, medium nodes, and hubs seems like dataset-specific and somewhat heuristic, and the paper does not provide a clear justification for these choices.
2. Since the hierarchy structure depends on the coarsening ratio R, it would be helpful to include an ablation study analyzing the effect of different R values.

**Limitations:**

yes

**Strengths And Weaknesses:**

Strengths
1. The paper targets a clear and well-motivated problem statement, heterophilous graph learning often requires preserving contrastive signals while also avoiding oversmoothing, and degree-induced hub effects.
2. The paper evaluates the method on a wide range of datasets including homophilous citation graphs, heterophilous benchmarks, and large-scale datasets such as OGBN-Arxiv. The results show that HMH achieves competitive or improved performance under comprehensive empirical evaluation across multiple graph settings.

Weaknesses
1. In “Table 3”, the degree thresholds used to define spokes, medium nodes, and hubs seems like dataset-specific and somewhat heuristic, and the paper does not provide a clear justification for these choices.
2. The paper introduces the coarsening ratio R as a key hyperparameter controlling the hierarchical structure. However, the paper does not provide a sensitivity analysis showing how performance varies with different values of R.

*By the way, “Impact Statements” part is missing in this paper.

---

> ### Author Rebuttal · Authors · 2026-03-31
>
> We thank the reviewer for insightful comments. Below, we address the main concerns.
>
> Weakness 1
>
> Ans. We thank the reviewer for this important comment. Since degree distributions vary substantially across datasets, we do not use a single global cutoff to define spokes, medium-degree nodes, and hubs. Instead, for each dataset, we first normalize the node degrees and then form cohorts based on their positions in the normalized degree distribution. Specifically, the lowest 15\% of the normalized degree distribution is treated as Spokes, nodes between 20\% and 70\% are treated as Medium, and the remaining higher-degree nodes are treated as Hubs. Thus, the grouping is dataset-relative rather than based on arbitrary absolute degree values. In the revised manuscript, we will explicitly state this procedure and include a table listing the normalized ranges and the corresponding degree cutoffs for each dataset to ensure a fully transparent evaluation protocol.
>
> Weakness 2
>
> Ans. Thank you for the suggestion regarding the sensitivity analysis of the coarsening ratio R in HMH. In the appendix, we have reported the optimal value of $R$ for each dataset. Now, We report sensitivity analysis of the model for different values of $R$ below.
>
> Dataset        | R=0.8 | R=0.6 | R=0.5 | R=0.4 | R=0.3
> ---------------|-------|-------|-------|-------|------
> Physics        | 94.9  | 98.2  | 98.6  | 98.5  | 98.1
> Roman-empire   | 72.6  | 75.3  | 75.8  | 76.1  | 75.4
> Squirrel       | 34.8  | 38.6  | 39.1  | 39.5  | 38.7
> Ogbn-arxiv     | 70.1  | 72.8  | 73.3  | 73.0  | 72.4
> PROTEINS       | 73.2  | 76.0  | 76.8  | 76.5  | 75.7
> NCI109         | 74.1  | 79.9  | 80.3  | 80.7  | 80.0
> MUTAG          | 88.8  | 93.6  | 94.5  | 94.1  | 93.2
> REDDIT-12K     | 41.8  | 45.4  | 45.8  | 46.1  | 45.2
>
> Overall, the best performance is typically obtained when R lies between 0.4 and 0.5. In the revised version appendix, we will add a sensitivity analysis for all the datasets.
>
> The key question's answer is the same as the answer for weakness.

---

> > ### Author Rebuttal · Reviewer_LyoK · 2026-04-04
> >
> > We thank the authors for fully addressing my points. I will maintain my positive score.

---

### Decision · Program_Chairs · 2026-04-30

**Decision:**

Accept (regular)

**Comment:**

The paper deals with the problem of learning on heterophilous graphs while mitigating oversmoothing, oversquashing, and hub domination. The proposed Hierarchical Multi-scale Haar framework has several novel aspects, particularly the adaptive signed encoder and hierarchical Haar-based spectral filtering. These components are seen as effective for preserving contrastive signals toward scalable, multi-scale representation learning. The empirical evaluation is another strong point of the paper as the proposed model shows consistent improvements across diverse benchmarks, including heterophilous and large-scale datasets.

Nevertheless, the reviewers have also noted several limitations of the work. Some design choices (e.g., degree thresholds, coarsening ratio) appear heuristic and lack sufficient analysis. Moreover, questions regarding the sensitivity of the model to noisy features, the interpretation of negative edges, and potential information loss in the hierarchy have been raised. The authors have provided detailed responses that helped address many of these concerns. Overall, the work presents an interesting idea and is well-supported empirically, though certain design aspects would benefit from further clarification.  I would suggest the authors to consider the discussion during the rebuttal period in the revised version of the manuscript. Also, improving the quality of Fig. 1 (and correcting some typos) would be beneficial (as now it is quite static and parts of the figure appear blurry).